# Semantic Mapping in Indoor Embodied AI – A Survey on Advances, Challenges, and Future Directions

**Sonia Raychaudhuri**                                                                *sraychau@sfu.ca*
*Simon Fraser University*

**Angel X. Chang**                                                                    *angelx@sfu.ca*
*Simon Fraser University*
*Alberta Machine Intelligence Institute (Amii)*

**Reviewed on OpenReview:** *https://openreview.net/forum?id=USgQ38RG6G*

## Abstract

Intelligent embodied agents (e.g. robots) need to perform complex semantic tasks in unfamiliar environments. Among many skills that the agents need to possess, building and maintaining a semantic map of the environment is most crucial in long-horizon tasks. A semantic map captures information about the environment in a structured way, allowing the agent to reference it for advanced reasoning throughout the task. While existing surveys in embodied AI focus on general advancements or specific tasks like navigation and manipulation, this paper provides a comprehensive review of semantic map-building approaches in embodied AI, specifically for indoor navigation. We categorize these approaches based on their structural representation (spatial *grids*, *topological* graphs, dense *geometric* or *hybrid* maps) and the type of information they encode (*implicit* features or *explicit* environmental data). We also explore the strengths and limitations of the map building techniques, highlight current challenges, and propose future research directions. We identify that the field is moving towards developing open-vocabulary, queryable, task-agnostic map representations, while high memory demands and computational inefficiency still remaining to be open challenges. This survey aims to guide current and future researchers in advancing semantic mapping techniques for embodied AI systems.

## 1 Introduction

Intelligent agents, whether physical robots or virtual embodied systems, must operate in complex, unstructured environments where effective behavior demands more than just low-level sensing and actuation. To act meaningfully, agents must form structured internal representations that link perception to reasoning and decision-making. Semantic maps serve this role by encoding not only spatial geometry but also high-level semantics of the environment (object categories, affordances, etc.). Semantic mapping is thus foundational in both robotics and embodied AI, particularly in open-world settings like autonomous driving (Bao et al., 2023; Li et al., 2024b), search-and-rescue operations (Gautham et al., 2023), automated cleaning robots (Singh et al., 2023) among others. While traditional mapping techniques prioritize geometric accuracy for localization and obstacle avoidance, recent progress in deep learning, computer vision and multi-modal perception has shifted focus toward semantically rich maps. Despite extensive literature, semantic mapping remains an open and rapidly evolving research area.

Existing surveys review semantic mapping within the context of its usage in downstream applications, thus largely centered on task advancement (Pfeifer & Iida, 2004; Kostavelis & Gasteratos, 2015; Cadena et al., 2017; Xia et al., 2020; Duan et al., 2022; Deitke et al., 2022; Zhu et al., 2021; Zhang et al., 2022b; Wu et al., 2024; Lin et al., 2024; Garg et al., 2020; Achour et al., 2022; Zheng et al., 2024a). Song et al. (2025) has recently reviewed semantic mapping in indoor robotics applications, but focused on the type of

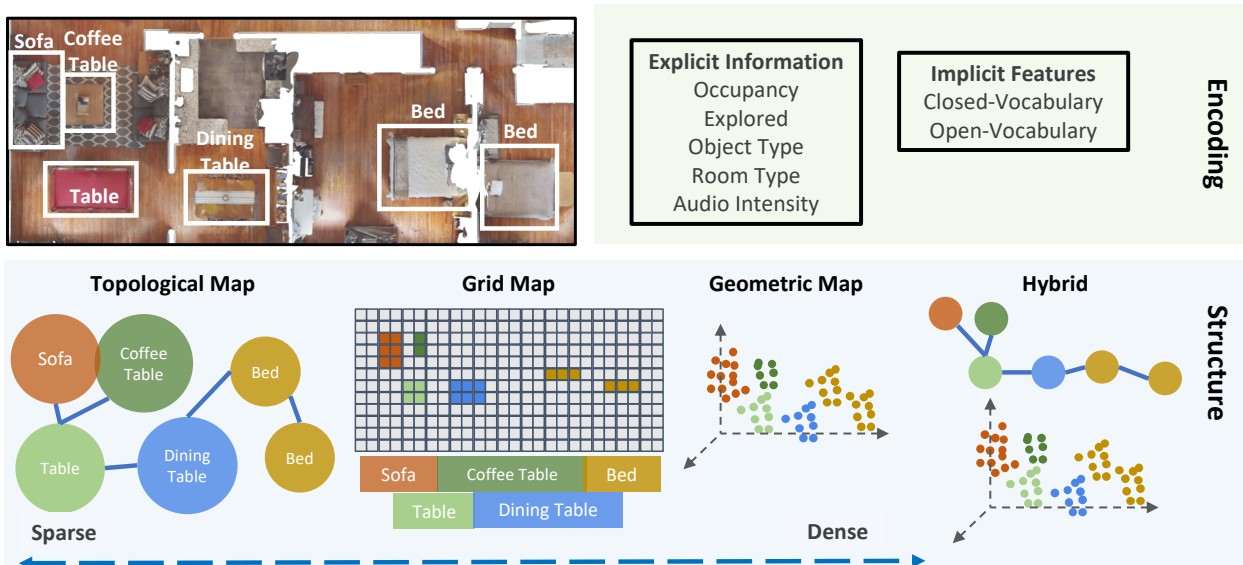

Figure 1: **Semantic maps.** The survey categorizes semantic map building methods in embodied agents based on their structure and the encoding it stores. *Structure*: a map of a physical environment can be structured as a topological map (with nodes and edges), a spatial grid, a dense geometric map or a hybrid map combining two or more of the others. *Encoding*: the structured maps can store either explicit (occupancy, object type, etc.) or implicit information (learned neural features) corresponding to the observation made at that location.

semantics acquired and the techniques of such extraction. In contrast, our survey offers a comprehensive review of semantic map-building methods, organized around the underlying map representations themselves, independent of specific downstream tasks. This perspective is both timely and necessary, given the recent advances in foundation models and the growing demand for general-purpose, open-vocabulary, task-agnostic representations.

To provide a principled understanding of semantic map-building methods, this survey categorizes the literature along two fundamental axes (see Fig. 1): map *structure* (e.g., topological graphs, spatial grids, dense geometric, and hybrid representations) and semantic *encoding* (explicit annotations vs. learned implicit features). This taxonomy reflects core design choices that influence map scalability, interpretability, generalization, multi-modal fusion and querying. By organizing approaches in this manner, we aim to unify diverse strands of research (see Tab. 1 for a summary of the reviewed papers), highlight the trade-offs between different representations and present key challenges and future opportunities in semantic mapping. This survey centers on semantic mapping approaches in the context of indoor mobile robots, a domain that offers a well-defined, practically relevant and technically rich environment for research. We focus on semantic mapping within embodied AI, that enables controlled environments capable of isolating high-level reasoning abilities of semantic maps from the complexities of noisy sensors and real-world observations. At the same time the survey establishes connections to SLAM-based approaches to underscore their methodological overlap and to bridge foundational techniques in robotics with emerging paradigms in embodied AI.

This survey is organized as follows. We begin the survey in Sec. 2 with background context to ground our discussion on semantic mapping. Sec. 3 introduces semantic maps and lays out the foundational components. We then explore different map structures in Sec. 4, followed by encoding strategies in Sec. 5. Sec. 6 reviews existing evaluation methodologies. In Sec. 7, we examine the key challenges in building semantic maps, leading into Sec. 8, which outlines promising directions for future research. We conclude in Sec. 9 by synthesizing the main insights and takeaways from the survey.

Table 1: **Summary table.** We characterize works that use maps for embodied AI as well as robotics by the type of structure they use (**Grid**, **Topological**, **Dense geometric maps**) and how information is encoded in the map (**Exp**licit vs **Imp**licit). *Explicit* encodings are pre-selected information such as occupancy $\boxed{\bullet}$, explored-area $\boxed{\text{x}}$, object category $\boxed{\triangle}$, visitation time $\boxed{t}$ and others. *Implicit* encodings are learned representations such as visual (V) or visual-and-language (VL) features. The use of VL features (typically from large pretrained models) enable building *open vocabulary* maps. Works that aggregate implicit features onto a grid map, but finally decode into explicit encodings are marked as 'Implicit to Explicit' in this table.

| Encoding | Structure | | |
|---|---|---|---|
| | **Grid** | **Topological** | **Dense geometric map** |
| Explicit (no semantics) | Occupancy Grids (Elfes, 1989) $\boxed{\bullet}$
ANS (Chaplot et al., 2019) $\boxed{\bullet}\,\boxed{\text{x}}$
UPEN (Georgakis et al., 2022a) $\boxed{\bullet}$
Integrating (Thrun et al., 1998b) $\boxed{\bullet}$
Combining (Tomatis et al., 2001) $\boxed{\bullet}$ | Topo SLAM (Choset & Nagatani, 2001)
Topomap (Blochliger et al., 2018)
Fast (Chen et al., 2022) | 6D-SLAM (Nüchter et al., 2007)
Robust 3D mapping (May et al., 2009)
Droid-SLAM (Teed & Deng, 2021)
Voldor+ SLAM (Min & Dunn, 2021) |
| Explicit (semantics) | Curious George (Meger et al., 2008) $\boxed{\bullet}\,\boxed{\text{x}}\,\boxed{\triangle}$
Efficient (Hu et al., 2013) $\boxed{\triangle}$
SemExp (Chaplot et al., 2020a) $\boxed{\bullet}\,\boxed{\text{x}}\,\boxed{\triangle}$
L2M (Georgakis et al., 2021) $\boxed{\bullet}\,\boxed{\triangle}$
SEAL (Chaplot et al., 2021) $\boxed{\bullet}\,\boxed{\triangle}$
CM² (Georgakis et al., 2022c)
MOPA (Raychaudhuri et al., 2023) $\boxed{\triangle}$
GOAT-Bench (Khanna et al., 2024) $\boxed{\bullet}\,\boxed{\text{x}}\,\boxed{\triangle}$
MapNav (Zhang et al., 2025) $\boxed{\bullet}\,\boxed{\text{x}}\,\boxed{\triangle}$
Instruction-guided (Wang et al., 2025) $\boxed{\triangle}$ | SLAM++ (Salas-Moreno et al., 2013) $\boxed{\triangle}$
Imitation (Duvallet et al., 2013)
Compact (Patki et al., 2019)
Multimodal (Arkin et al., 2020a)
LIFGIF (Raychaudhuri et al., 2025) $\boxed{\triangle}$
ASHiTA (Chang et al., 2025) | Towards semantic maps (Nüchter & Hertzberg, 2008) $\boxed{\triangle}$
3D object-class map (Stückler et al., 2012) $\boxed{\triangle}$
Parsing (Triebel et al., 2012) $\boxed{\triangle}$
Street Scenes (Floros & Leibe, 2012) $\boxed{\triangle}$
Dense 3D (Hermans et al., 2014) $\boxed{\triangle}$
SemanticFusion (McCormac et al., 2017) $\boxed{\triangle}$ |
| | Following directions (Matuszek et al., 2010)
Conceptual (Zender et al., 2008)
SemanticGraph (Hemachandra et al., 2014)
Learning semantic maps (Walter et al., 2014)
Learning models (Hemachandra et al., 2015)
Inferring maps (Duvallet et al., 2016)
BEVBert (An et al., 2023) $\boxed{\bullet}\,\boxed{\triangle}$
LaneSegNet (Li et al., 2024b) | | |
| | | Contextually guided (Anand et al., 2013) $\boxed{\triangle}$
3DSG (Armeni et al., 2019) | |
| | | 3D-DSG (Rosinol et al., 2020a) $\boxed{\bullet}\,\boxed{\triangle}$
Hydra (Hughes et al., 2022) $\boxed{\bullet}\,\boxed{\triangle}$
S-graphs (Shen et al., 2019) $\boxed{\bullet}\,\boxed{\triangle}$
S-graphs+ (Bavle et al., 2023) $\boxed{\bullet}\,\boxed{\triangle}$
CURB-SG (Greve et al., 2024) $\boxed{\bullet}\,\boxed{\triangle}$ | |
| Implicit to Explicit | SemanticMapNet (Cartillier et al., 2021) $\boxed{\triangle}$ | SceneGraphFusion (Wu et al., 2021) | |
| Implicit (V) | CMP (Gupta et al., 2017)
MapNet (Henriques & Vedaldi, 2018)
MultiON (Wani et al., 2020)
RNR-Map (Kwon et al., 2023) | SPTM (Savinov et al., 2018)
NTS (Chaplot et al., 2020b)
CMTP (Chen et al., 2021)
VGM (Kwon et al., 2021) $\boxed{t}$ | KPConv (Thomas et al., 2019)
PointResNet (Desai et al., 2022)
Neural Fusion (Mazur et al., 2023) |
| Implicit (VL) | CoW (Gadre et al., 2023)
VLMap (Huang et al., 2023a)
Le-RNR-Map (Taioli et al., 2023)
NLMap (Chen et al., 2023a)
VLFM (Yokoyama et al., 2023)
VoxPoser (Huang et al., 2023b)
InstructNav (Long et al., 2024)
OVL-MAP (Wen et al., 2025) | LM-Nav (Shah et al., 2023)
RoboHop (Garg et al., 2024)
Clio (Maggio et al., 2024) | OpenScene (Peng et al., 2023)
CLIPFields (Shafiullah et al., 2023)
ConceptFusion (Jatavallabhula et al., 2023)
CLIP2Scene (Chen et al., 2023c)
3D aware ObjNav (Zhang et al., 2023a) |
| | | ConceptGraphs (Gu et al., 2024)
HOV-SG (Werby et al., 2024) | |
| | | StructNav (Chen et al., 2023b) | |

# 2 Background Reading

In this section, we provide background readings that help to contextualize the survey on semantic mapping. We first introduce key tasks (Sec. 2.1) in both robotics (Sec. 2.1.1) and embodied AI (Sec. 2.1.2), since semantic mapping is often studied in conjunction with such tasks. In Sec. 2.2, we discuss the basics (Sec. 2.2.1) and techniques of SLAM (Sec. 2.2.2) as well as the historic evolution of Semantic SLAM (Sec. 2.2.3), which has an overlap with the map building techniques that we later discuss throughout the survey. Next we briefly discuss two popular system design choices in Sec. 2.3, end-to-end learning (Sec. 2.3.1) and modular pipeline (Sec. 2.3.2), since it helps the readers understand how semantic map construction varies between end-to-end and modular approaches.

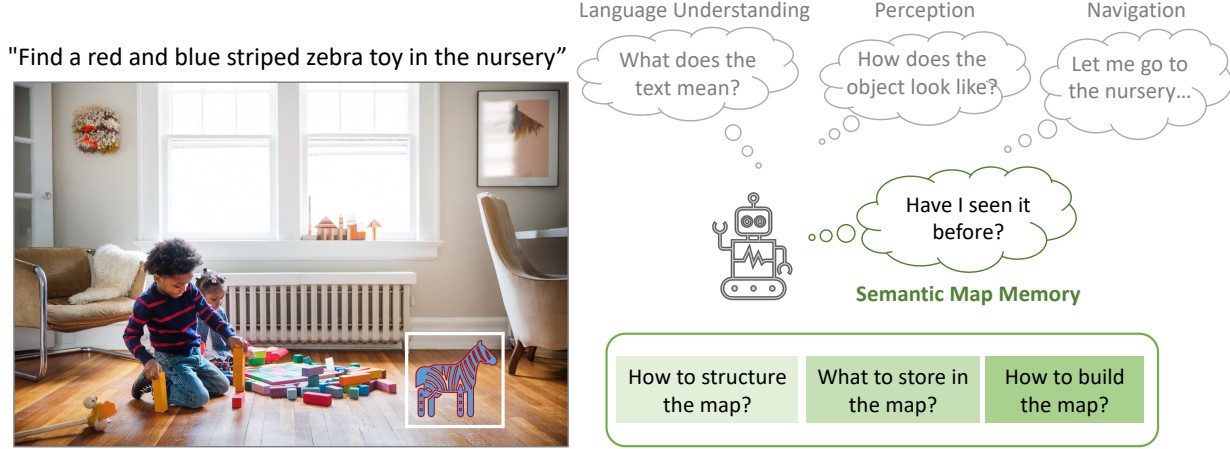

Figure 2: **Motivation.** To perform a complex task in an indoor environment, the robotic agent must possess multiple skills of language understanding, visual perception, navigation, etc. Among these the most crucial is building and maintaining a semantic map of the environment so that it can come back to it while performing the task.

## 2.1 Embodied tasks

Embodied tasks involve an intelligent agent, either a physical robot or a simulated body, perceiving and interacting with the environment through its embodiment (sensors, actuators etc.). These tasks require the agent to not only understand the world (via vision, language, etc.) but also take meaningful actions within it (such as navigation or object manipulation). In this section, we provide a brief overview of the embodied tasks explored in both robotics and embodied AI, highlighting their evolution as a growing research direction. This serves to contextualize our focus on semantic mapping, which is often studied in conjunction with such embodied tasks.

### 2.1.1 Robotics tasks

The era of modern robotics began with Unimate (Detesan & Moholea, 2024), the first industrial manipulator arm and Shakey (Center, 1984), the first mobile robot. Since then the field has evolved to have increased autonomy and complex reasoning skills. This has been enabled by research in various areas spanning basic navigation and obstacle avoidance to complex real-world capabilities involving perception, mapping, and manipulation. Early research focus on collision avoidance (Fox et al., 1997), Monte Carlo Localization (Thrun et al., 2001) and SLAM frameworks enabling robots to localize while mapping unknown environments (Thrun, 2002; Thrun & Montemerlo, 2006; Taheri & Xia, 2021). As sensors have improved over the years, semantic mapping gained traction, integrating object recognition and scene understanding into spatial maps (Salas-Moreno et al., 2013). Concurrently, robotic manipulation has become a central focus, with tasks such as pick-and-place, insertion, and rearrangement. Classical approaches like force-control-based manipulation (Raibert & Craig, 1981; Yoshikawa, 1985) have been complemented by modern reinforcement learning-based systems like QT-Opt (Kalashnikov et al., 2018) and Transporter Networks (Zeng et al., 2020). The field has also expanded to multi-task and long-horizon planning, where agents are required to complete sequences of interconnected tasks rather than isolated actions. A prominent framework in this area is Task and Motion Planning (TAMP) (Garrett et al., 2020), which integrates high-level symbolic reasoning (e.g., planning to "pick up the cup, open the door, and place the cup on the table") with low-level continuous motion planning (e.g., calculating joint angles and trajectories to execute those actions). This enables agents to operate in complex and dynamic environments. Recent trends include uncertainty-aware planning (Stachniss et al., 2005; Blanco et al., 2008a; Georgakis et al., 2022a; Carlone et al., 2013; Pan et al., 2019), semantic mapping (Rosinol et al., 2020b) and task planning in dynamic environments (Rosinol et al., 2020a). Another fast growing field in robotics is autonomous driving (Guan et al., 2024; Zhao et al., 2024), where recent trends increasingly rely on Bird's Eye View (BEV) representations, which transform multi-view sensor inputs into a unified top-down map

using end-to-end learned models. BEV is a common intermediate representation used to simplify reasoning about spatial layouts, obstacles, lanes and other semantic elements. Transformer-based architectures like BEVFormer (Li et al., 2024c) and BEVFusion (Liu et al., 2023b) have emerged as state-of-the-art, enabling spatially consistent, semantic-rich BEV maps that support tasks such as detection, planning and trajectory prediction. For example, LaneSegNet (Li et al., 2024b) trains an end-to-end system to predict lane segments from multi-view surrounding images. Unlike semantic maps that are discussed throughout this survey, BEV maps are typically short-lived local (per-frame) representations and specialized for driving scenarios. Overall, the advances in robotics collectively demonstrate a shift from reactive, single-task robots toward robust, general-purpose systems capable of operating autonomously in dynamic real-world environments.

### 2.1.2 Embodied AI tasks

The field of embodied AI began to emerge prominently around 2017–2018, catalyzed by the availability of simulation environments, such as AI2-THOR (Kolve et al., 2017), Habitat (Savva et al., 2019), etc., and benchmarks such as Vision-and-Language Navigation (VLN) (Anderson et al., 2018b), Embodied Question Answering (EQA) (Wijmans et al., 2019a), etc. Historically, embodied AI diverged from robotics by focusing more heavily on learning-based agents interacting with simulated environments using vision, language, and action, often without relying on physical hardware. The divergence was motivated by the scalability and reproducibility challenges in real-world robotics, and the need to study cognition, planning, and multi-modal learning at scale. Broadly speaking, embodied AI positions itself at the intersection of computer vision, natural language processing and reinforcement learning, while classical robotics focuses on control, perception and physical interaction with the real world.

Embodied AI tasks vary depending on the type interaction of an agent with its environment. Broadly, we can group embodied tasks into three groups – *Exploration* task (Chaplot et al., 2019) requires an agent to efficiently explore its environments; *Navigation* task (Wijmans et al., 2019b; Batra et al., 2020b) requires the agent to take actions in order to move around the environment; *Manipulation* task (Szot et al., 2021; Weihs et al., 2021) requires the agent to perform interactive actions to change the state of other objects in the environment. The taxonomy of tasks can be further broken down by the target specification provided to the agent, which impacts the information that need to be retained. For instance, for navigation, the following are commonly studied. In *Point-Goal Navigation* (Wijmans et al., 2019b) (PointNav), the agent is given a target coordinate relative to its starting position, whereas in *Image-Goal Navigation* (ImageNav) it is given a target image (Chaplot et al., 2020b). In the *Object-Goal Navigation* (ObjectNav) task, the agent needs to navigate to any instance of an object category (Yadav et al., 2022). An extension to the ObjectNav task is the *Multi-Object Navigation* (MultiON) (Wani et al., 2020) task where the agent is required to navigate to multiple objects in a particular sequence. *Vision-and-Language Navigation* (VLN) (Anderson et al., 2018b) requires the agent to find the target as specified by a natural language instruction. In *Audio-Visual Navigation* task, the agent needs to navigate to an object emitting a particular sound in an indoor environment (Chen et al., 2020a; Gan et al., 2019). Depending on the type of task, it may be sufficient to store just the object category (e.g. for ObjectNav) in the map, or it is may be necessarily to retain more finer-grained information (e.g. VLN). In this survey, we will mainly focus on recent work on room-scale map building for navigation as these methods can be extended for maps for manipulation and used for exploration.

## 2.2 Simultaneous Localization and Mapping (SLAM)

Although we briefly discuss robotics applications in Sec. 2.1, we address Simultaneous Localization and Mapping (SLAM) (Thrun et al., 1998a; 2000; Ferris et al., 2007; Huang et al., 2011; Carlone et al., 2024) separately here due to its close connection with semantic mapping, particularly the Semantic SLAM literature. While both aim to enrich spatial representations with semantic information, they differ in scope and purpose. Rooted in robotics, Semantic SLAM focuses on building globally consistent, pose-aware maps augmented with semantics to improve localization, mapping accuracy and long-term robustness. In contrast, semantic mapping in embodied AI emphasizes representations that support high-level reasoning and decision-making, often abstracting away precise localization and low-level sensor noise in favor of adaptability.

### 2.2.1 Basics of SLAM

SLAM (Simultaneous Localization and Mapping) enables a robot to leverage data from multiple sensors (cameras, LiDAR, IMU etc.) to perceive the environment and build a map while simultaneously localizing itself on the map. Mathematically speaking, SLAM aims to estimate the robot's trajectory and a map of the environment simultaneously, using noisy sensor data and imperfect motion estimates. A robot's motion is modeled by a state transition function that predicts its next pose based on the previous pose and control inputs:

$$x_t = f(x_{t-1}, u_t) + w_t \tag{1}$$

where, $x_t$ is the robot's state at time $t$, $u_t$ is the control input (such as wheel odometry or IMU), and $w_t$ is process noise. Simultaneously, the robot makes observations of the landmarks in the environment, which is modeled by the measurement equation:

$$z_t^i = h(x_t, m_i) + v_t \tag{2}$$

where $z_t^i$ is the observation of landmark $i$ at time $t$, $m_i$ is the position of the landmark and $v_t$ is the measurement noise. The difference between the predicted and the actual observations is the residual (error):

$$e_t(x) = z_t^i - h(x_t, m_i) \tag{3}$$

SLAM formulates the estimation problem as a nonlinear least-squares optimization, which aims at finding the optimal trajectory $x$* and landmark positions $m$* that best explain all measurements:

$$x^*, m^* = arg \ \min_{x,m} \sum_t \sum_i e_t^T(x) \ \Omega_t e_t(x) \tag{4}$$

where $\Omega_t$ is the information matrix (inverse covariance) that weights the measurements according to their uncertainty, i.e. more weight is given to a measurement that is very certain. $e_t^T(x)$ is the transpose of the error $e_t(x)$ and the term $e_t^T(x) \ \Omega_t e_t(x)$ represents weighted squared error. In graph-based SLAM, this optimization corresponds to optimizing a factor graph (Loeliger, 2004; Dellaert et al., 2017), where nodes represent robot poses and edges represent spatial constraints, such that Equation (4) becomes:

$$x^* = arg \ \min_x \sum_{(i,j) \in \varepsilon} ||z_{ij} - \hat{z}_{ij}(x_i, x_j)||_{\Omega_{ij}}^2 \tag{5}$$

where, $(i, j) \in \varepsilon$ represent edges in the pose graph, $z_{ij}$ is the relative pose measurement between nodes $i$ and $j$ , $\hat{z}_{ij}$ is the predicted relative measurement from current estimates, and $|| \ . \ ||_{\Omega_{ij}}^2$ is the Mahalanobis distance (weighted squared error). This global optimization ensures that the estimated map and trajectory are as consistent as possible with the entire history of noisy sensor data.

### 2.2.2 SLAM Techniques

Active SLAM (Ahmed et al., 2023) enables an autonomous agent to actively choose its actions to improve its map and localization, rather than passively mapping the environment as it moves (Fig. 3-left). The SLAM system (Cadena et al., 2016; Pu et al., 2023) primarily consists of a front-end module and a back-end module (Fig. 3-right). The front-end module computes feature extraction, data association and feature classification on the sensor observations. Moreover it applies algorithms such as Iterative Closest Point (ICP) (He et al., 2017b) that aligns sensor observations from subsequent frames into consistent 3D geometry as well as loop closure (Tsintotas et al., 2022) that recognizes whether the current observation matches with a previously visited area. The backend module, on the other hand, is responsible for pose optimization and map estimation based on the data from the front-end. The back-end also provides feedback to the front-end for loop closure and verification. While some SLAM methods use LiDARs (Khan et al., 2021), others use camera as the primary sensor (visual SLAM or vSLAM), due to the availability of less expensive cameras and the advances in computer vision. Visual SLAM methods often extract and match geometric features (points, lines, or planes) (Mur-Artal et al., 2015; Yang & Scherer, 2017; Kaess, 2015; Cai et al., 2021) from the image, with the help of feature detection algorithms such as SIFT (Lowe, 2004), SURF (Bay et al., 2008), and ORB (Rublee

et al., 2011). Others operate directly on the image pixel intensities (Newcombe et al., 2011b; Engel et al., 2014) and thus retain all information about the image. While these methods use RGB camera as the sensor, RGB-D SLAM (Newcombe et al., 2011a; Kaess et al., 2012; Endres et al., 2013; Whelan et al., 2016) uses RGB-D cameras to simultaneously collect color images as well as depth images. In contrast, Visual-Inertial SLAM (Mur-Artal & Tardós, 2017; Cheng et al., 2021) uses an additional IMU sensor to mitigate the effects of image blur and poor illumination from using camera sensor alone.

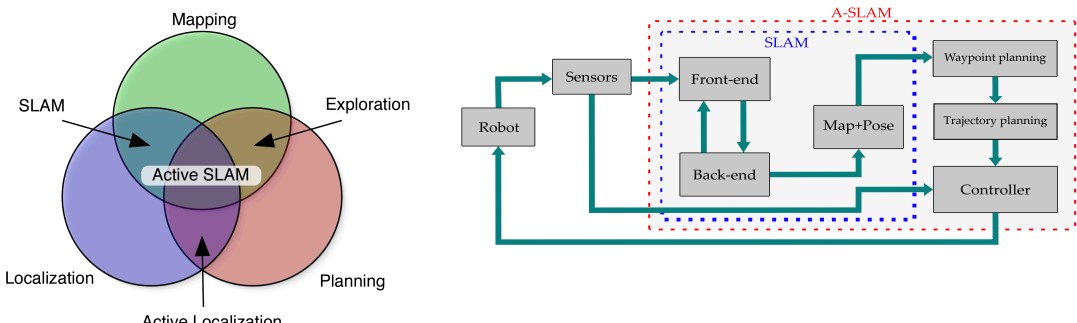

Figure 3: **SLAM.** (left) At the core of classical mobile robotics lie three core tasks – mapping, localization, and planning. These are often interdependent on each other and overlap to form other tasks such as SLAM (localization and mapping), exploration (mapping and planning), active localization (planning and localization) and active SLAM (mapping, localization, and planning) [*figure reproduced from Fairfield (2009)*]. (right) Architecture of SLAM includes the front-end to handle perception tasks and the back-end to estimate pose and the map. Active SLAM (A-SLAM) additionally includes planning and controller modules [*Figure reproduced from Ahmed et al. (2023)*].

### 2.2.3 Semantic SLAM

While geometric maps in SLAM systems prove to be effective for simple navigation and control (Shan et al., 2020b; Campos et al., 2021), they provide limited understanding of the environment's semantic content. As robotic systems grew more complex and began interacting with unseen unstructured dynamic environments, the need for higher-level understanding led to the emergence of Semantic SLAM. It enriches spatial geometric maps with meaningful concepts like objects, rooms, and affordances, thus bridging the gap between perception and task-level reasoning in modern embodied agents. Early approaches do so by matching image features to object model databases (Civera et al., 2011; Salas-Moreno et al., 2013) or by incorporating semantic and geometric cues into structure-from-motion (Bao & Savarese, 2011; Bao et al., 2012) and SLAM optimization processes (Fioraio & Di Stefano, 2013). With advances in deep learning, recent works on Semantic SLAM (McCormac et al., 2017; Yin et al., 2020) have improved semantic representation, real-time performances and dynamic environment modeling. Some works show that using semantic maps improves the performance of the various modules in a SLAM system (Xiang & Fox, 2017; Tateno et al., 2017; Qin et al., 2021; Qian et al., 2021), while others show that using SLAM to impose consistency constraints improves semantic segmentation (Mozos et al., 2007; Lai et al., 2014; Pronobis & Jensfelt, 2012; Pillai & Leonard, 2015; Cadena et al., 2015). Many also show that combining SLAM and semantic segmentation into joint frameworks enhances both mapping and object recognition (Flint et al., 2011; Bao et al., 2012; Hane et al., 2013; Kundu et al., 2014; Sengupta & Sturgess, 2015; Vineet et al., 2015).

Moreover, semantic SLAM works can be categorized based on the structure and encoding of the semantic maps, in the same way as semantic mapping in embodied AI. For example, some works use sparse representations (Bowman et al., 2017; Yang & Scherer, 2019; Nicholson et al., 2018; Chen et al., 2020d; Shan et al., 2020a; Atanasov et al., 2018; Feng et al., 2019; Salas-Moreno et al., 2013), while others use dense

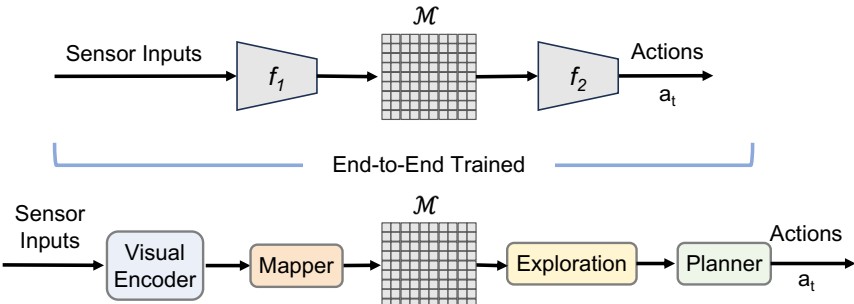

Figure 4: **End-to-end vs Modular.** (Top) End-to-end model is trained as a single pipeline which generates actions directly from sensory inputs. (Bottom) Modular pipeline consists of various sub-modules, each with a specific function so that they can be trained independently of the others.

representations (Rosinol et al., 2021; Grinvald et al., 2019; McCormac et al., 2018; Chen et al., 2019b; Maturana et al., 2018; Miller et al., 2021; 2022). Some works store explicit semantic concepts such as rooms (Pronobis & Jensfelt, 2012) or objects (Pillai & Leonard, 2015) in the map, while others store implicit neural features (Xiang & Fox, 2017; Tateno et al., 2017; Qin et al., 2021; Shah et al., 2023; Maggio et al., 2024; Werby et al., 2024). Throughout this survey, we will revisit these papers to categorize them based on the categories we use for semantic mapping in embodied AI.

For a more comprehensive reading on SLAM and semantic SLAM, we refer the readers to the following survey papers – Thrun (2003); Kostavelis & Gasteratos (2015); Cadena et al. (2017); Younes et al. (2017); Taketomi et al. (2017); Landsiedel et al. (2017); Saputra et al. (2018); Sualeh & Kim (2019); Chen et al. (2020c); Lluvia et al. (2021); Taheri & Xia (2021); Tourani et al. (2022); Pu et al. (2023); Racinskis et al. (2023); Sousa et al. (2023); Wang et al. (2024); Chen et al. (2025).

## 2.3 System design strategies

Designing systems for embodied agents involves a fundamental architectural choice between end-to-end learning and modular pipelines. While end-to-end approaches map raw sensory input directly to actions using a single neural network, modular systems decompose the task into interpretable components. Understanding this distinction is essential for situating semantic mapping within the broader system design, as it influences how maps are constructed, represented and utilized. This section briefly outlines these paradigms (Sec. 2.3.1 and Sec. 2.3.2) and discuss trade-offs between them in Sec. 2.3.3, to provide a background for the mapping methods discussed in the remainder of this survey.

### 2.3.1 End-to-end approaches

The embodied AI and robotics community has seen a lot of progress in training task-specific end-to-end models with reinforcement learning (RL) (Tai et al., 2017) that directly learns to predict discrete (Wani et al., 2020) or continuous actions (Kalapos et al., 2020) from visual observations (see Fig. 4). These methods may consist of an unstructured memory such as LSTM (Mei et al., 2016) (Dobrevski & Skočaj, 2021). However, such representation lacks reasoning about 3D space and geometry and fail to perform well in long-horizon path planning. This has lead to the development of approaches that build an intermediate map representation. Such a map can be implemented using differentiable operations so as to facilitate end-to-end training. Gupta et al. (2017) shows that an egocentric map built this way is beneficial for both PointNav and ObjectNav tasks, whereas Henriques & Vedaldi (2018) learns a global map for the task of localization. While these methods are trained using supervised learning, Wani et al. (2020) use RL to learn to predict actions based on an intermediate global map of the environment to address the complex MultiON task. In autonomous driving and planning, end-to-end approaches aim to learn a direct mapping from raw sensor inputs (e.g., images, LiDAR) to control outputs or motion trajectories. These methods often leverage transformer-based architectures (Lin et al., 2022) to jointly model perception, spatial reasoning and decision-making within a single framework. By optimizing the entire pipeline holistically, they promise improved robustness and

efficiency, particularly in complex, dynamic environments. Recent work demonstrates that such end-to-end models can learn to implicitly reason about obstacles, goals, and traffic patterns. However challenges remain in interpretability, generalization and safety-critical validation. However, irrespective of the map representation or the mode of training, these approaches need to be retrained every time the task definition changes and even the basic skills required to perform a task need to be learned from scratch.

### 2.3.2   Modular pipeline

Another line of work explores how to breakdown a complex navigation task into a set of basic skills that the agent needs to acquire. Such skills can then be learned independently of each other so that they can be leveraged across various tasks without the need to be retrained from scratch. While some use two modules (Bansal et al., 2020), one for high-level decision making and another for low-level planner, others (Chaplot et al., 2019; 2020a; Gervet et al., 2023; Raychaudhuri et al., 2023) further breakdown into sub-modules, such as visual encoder, mapper and exploration in addition to the low-level planner. Fig. 4 shows the difference between the end-to-end approach and the module approach, and we elaborate on the modules and popular design choices below.

**Visual Encoder.** This module encodes agent observations to produce semantic visual features and predictions at every time step. Prior works have used visual features from pretrained backbones such as ResNet He et al. (2016) or ViT (Dosovitskiy et al., 2020), and often leveraging object detectors MaskRCNN He et al. (2017a) or FasterRCNN Ren et al. (2015). As we will see in Sec. 5.2.2, with the development of large pretrained vision-language models and open-vocabulary detectors, pretrained models such as CLIP (Radford et al., 2021), LSeg (Li et al., 2022), DINO (Caron et al., 2021; Oquab et al., 2023), and others are increasingly popular as the basis for building large to open-vocabulary maps. The visual encoder used will determine the information captured in the features, and whether there are detected object instance bounding boxes or segmentations for integration into the mapping modules.

**Mapper.** The mapper is responsible for building a semantic map of the environment from the encoded image features and agent pose. To build a global map over time, the mapper typically aggregates the current map with the map from previous step (see Sec. 3.4 for details). In this paper, we survey how recent methods structure the map (Sec. 4) and what information can be encoded in it (Sec. 5). Tab. 2 summarizes the type of information stored in the map by various methods. This can be occupancy information, explored area or semantic labels of the detected objects.

**Exploration.** This module enables the agent to explore its environment efficiently to either ensure the map is complete (by maximizing the covered area) or selecting unvisited areas where the target is likely to be. Typically, the exploration module selects a point or region to explore given the obstacle map built by the mapper and the current agent pose. Agents can use simple heuristics-based methods such as sampling a point at uniform (Zhang et al., 2021; Raychaudhuri et al., 2023), systematically sampling four corners of a grid centered at the agent (Luo et al., 2022) or selecting a point from the unexplored frontier (Yamauchi, 1997). To decide which frontier point the agent should explore, various strategies are employed, such as selecting the nearest point to the agent (Gervet et al., 2023) or the most promising point based on semantic reasoning. In the semantic reasoning based exploration methods, the agents may select the highest text-image relevance score (Gadre et al., 2023; Yokoyama et al., 2023) from a pretrained large vision-language model such as BLIP-2 (Li et al., 2023), select the highest probabilistic output of the VLM directly (Ren et al., 2024), or leverage a LLM to extract common-sense knowledge (Zhou et al., 2023). Researchers have also used learned policies (Chaplot et al., 2019; 2020a), where the agents are generally trained with RL using rewards, such as coverage (Chen et al., 2019a) or curiosity (Pathak et al., 2017; Mazzaglia et al., 2022). Although there is less hand-crafted rules in learning-based methods, they need millions of training steps and careful reward engineering.

**Planner.** Once a map is built, a low level path-planning module is used to plan a path to the goal location from the agent's current location. The path consists of low-level actions that can be executed by the agent to move to the goal. While this is implemented as a heuristics-based Fast Marching Method (Sethian, 1996) in most of the prior works, a recent approach MOPA by (Raychaudhuri et al., 2023) has used a learned PointNav policy trained offline with DD-PPO (Wijmans et al., 2019b).

In Tab. 2, we compare common modular approaches and look at how they leverage various heuristics-based or learned approach for each of the modules. The advantages of a modular pipeline include its ability to leverage pretrained models from other tasks (Gervet et al., 2023; Raychaudhuri et al., 2023) and its ability to transfer from simulation to real-world robots better (Gervet et al., 2023).

Table 2: We show how prior works build map either as part of an end-to-end architecture or as a modular architecture consisting of four basic modules. For the modular approaches, we summarize the different module choices as well as what they store in their map.

| Methods | Task | End-to-End | Modular | | | |
| | | | Visual Encoder | Map | Exploration | Planner |
| --- | --- | --- | --- | --- | --- | --- |
| ANS (Chaplot et al., 2019) | Exploration | | ResNet18 (He et al., 2016) | occupancy + explored | learned policy | Fast Marching |
| NTS (Chaplot et al., 2020b) | ImageNav | | ResNet18 (He et al., 2016) | topological map | learned policy | A* |
| SemExp (Chaplot et al., 2020a) | ObjectNav | | MaskR-CNN (He et al., 2017a) | occupancy + explored + semantic labels | learned policy | Fast Marching |
| ModLearn (Gervet et al., 2023) | ObjectNav | | MaskR-CNN (He et al., 2017a) | occupancy + explored + semantic labels | learned SemExp | Fast Marching |
| MOPA (Raychaudhuri et al., 2023) | MultiON | | FasterRCNN (Ren et al., 2015) | semantic labels | Uniform Sampling Exploration | PointNav(Wijmans et al., 2019b) |
| CMP (Gupta et al., 2017) | PointNav, ObjectNav | ✓ | | | | |
| MapNet (Henriques & Vedaldi, 2018) | Localization | ✓ | | – | | |
| MultiON (Wani et al., 2020) | MultiON | ✓ | | | | |

### 2.3.3 Trade-offs

Both the design choices have their own merits and limitations. End-to-end systems are simple to train and deploy, and they directly optimize task performance. However, they often lack interpretability, generalize poorly to new scenarios, and make component reuse difficult. In contrast, modular pipelines offer greater transparency, reusability, and flexibility to combine learned and classical methods, making them well-suited for complex tasks involving long-horizon planning and semantic reasoning, despite the potential for error propagation and suboptimal integration of modules.

## 3 Semantic map

In this section, we introduce semantic maps (Sec. 3.1), and provide an overview of their structural representations (Sec. 3.2), semantic encodings (Sec. 3.3), and the process of building them (Sec. 3.4).

### 3.1 What are semantic maps?

While maps, capturing the geometry of the space, can help an agent avoid obstacles, they are not sufficient for the demands of more complex reasoning tasks. An enhanced map that goes beyond geometry to capture meaning and context in its environment aligns with how humans perceive and navigate its surroundings. These are *semantic maps*, which provide a richer and more nuanced understanding about the objects and places in the environment and are indispensable for performing complex tasks such as navigating to a specific room (kitchen) (Narasimhan et al., 2020), rearranging objects (Trabucco et al., 2022) or performing an action on a specific object (sitting on a couch) (Peng et al., 2023). A physical or a virtual agent perceives the environment through sensors (camera, LiDAR, etc.), uses cognition to classify the objects and regions it perceives (Ren et al., 2015; He et al., 2017a; Liu et al., 2024; Kirillov et al., 2023; Zhang et al., 2023b), stores them in a structured semantic map which can be queried for efficient reasoning.

### 3.2 What is the structure of this map?

A semantic map can be structured as a *spatial grid map*, *topological map*, *dense geometric* or a *hybrid map* (Fig. 1). A spatial grid map is a top-down grid where each grid cell represents an area in the physical environment. So if an object is at a certain location $(X, Y, Z)$ in a 3D scene, the semantic map will contain information about that object at the corresponding grid cell $(x, y)$, where $x$ and $y$ are the row and column numbers respectively such that there is a direct mapping from $(X, Y, Z)$ to $(x, y)$. For navigation, most spatial

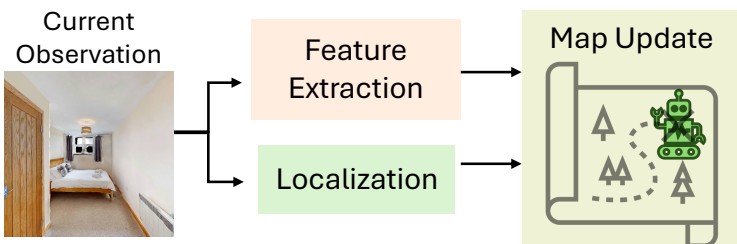

Figure 5: Map building involves *localization* (where the agent is on the map), *feature extraction* (extracting useful semantic information from the observations), and *map update* (building the map by aggregating the semantic information over time).

maps are 2D, such that a grid cell ignores the $Z$-axis (up direction) by aggregating the semantic information across the up axis. However they can also be 3D where the $Z$-axis is divided into discrete bins. On the other hand, a topological map is a graph-like structure where nodes represent objects or important landmarks in the scene and edges represent relationship (distance, spatial relation, etc.) between them. It is also possible to store semantic information on a dense geometric, which can be viewed as a 3D map with varying density. In a dense geometric, information is associated with each point $(x, y, z)$ corresponding to 3D location $(X, Y, Z)$ in the physical space. Unlike the voxel-grid, which is regularly spaced, points can be sampled at varying densities. Some works combine two or more of the above structures to form a hybrid map since each structure has its own advantages and limitations. We discuss each of these in detail in Sec. 4.

### 3.3   What encoding is stored in this map?

The semantic map stores information about a particular 3D location $(X, Y, Z)$ in the physical environment. This information can either be *explicit* or *implicit*. Explicit encodings have clear specific meanings assigned to each value. For instance, each cell $(X, Y, Z)$ can store information about whether there are any obstacles at that position, whether that location has been explored by the agent, the category of the object present there and so on. On the other hand, an implicit encoding is a feature encoding capturing information derived from the sensory input (e.g. images) that the agent observes at that particular location $(X, Y, Z)$. The features are typically extracted from pre-trained encoders. Depending on whether the feature encoder was pre-trained on a set of images from limited categories or a large internet-scale dataset of image and language data, the implicit encoding can be either *closed-vocabulary* or *open-vocabulary*. The term *closed-vocabulary* is used to indicate only a limited set of object categories is recognized, while in a *open-vocabulary* setting, the features extractors can theoretically identify 'any' object.

### 3.4   How is the map built?

Creating accurate and detailed semantic maps requires integrating data from various sources and sensors such as camera, LiDAR and depth sensors. More specifically, map building consists of having an agent navigate about a space, and accumulating observations $O_t$ at time step $t$ into the appropriate map structure $m_t$. To build an accurate map, the agent first needs to have an estimate of where it is (*localization*). Next it extracts semantic information from an observation $F(O_t)$ (*feature extraction*), and combines the features into a common map over time (*accumulation*) (refer to Fig. 5). While building a spatial grid map, an additional step is to project the features onto the map (*projection*). It is common to group the last three steps into *map building* and study it jointly with *localization* in Simultaneous Localization and Mapping (SLAM). We discuss SLAM methods briefly in Sec. 2.2.

**Localization.** Localization can be challenging due to noisy sensors and actuators. To simplify the problem, it is common in the embodied AI community to either assume perfect localization is given at each time step (Cartillier et al., 2021) or to localize the agent with respect to its starting position in an episode (Henriques & Vedaldi, 2018) assuming perfect actuation. The latter is more easily adapted to the real-world setting since

it doesn't require the exact knowledge about the agent's pose. Instead the relative displacement of the agent with respect to its starting pose is enough to build the map eventually.

**Feature extraction.** Feature extraction is a crucial part of building a semantic map. Ideally these features should be representative of the objects present in the map. We discuss this topic at length in section Sec. 5.

**Projection.** An important step in building a spatial grid map is taking the 2D observations and project them into 3D. Typically, this relies on having depth information and known camera parameters in order to convert 2D pixel coordinates to 3D world coordinates. To project a particular pixel in the camera frame, first a ray is shot from the camera center through the image pixel $(i,j)$ to the depth $d_{i,j}$ to get a 3D point in the camera coordinate frame. Next the camera coordinates are converted to the world coordinates $(X, Y, Z)$. For a 2D spatial map, the 3D coordinate$(X, Y, Z)$ is mapped to the grid cell indices $x$ and $y$ in the spatial map. The transformation for the standard pinhole camera with known camera pose (3D rotation $R$ and 3D translation $t$) and intrinsics $(K)$ can be written as:

$$\begin{bmatrix} X \\ Y \\ Z \end{bmatrix} = d_{i,j} R^{-1} K^{-1} \begin{bmatrix} i \\ j \\ 1 \end{bmatrix} - t \tag{6}$$

and the orthographic projection can be written as,

$$\begin{bmatrix} x \\ y \\ 0 \\ 1 \end{bmatrix} = P_v \begin{bmatrix} X \\ Y \\ Z \\ 1 \end{bmatrix} \tag{7}$$

where where $P_v$ is known orthographic projection matrix to convert 3D world coordinates into 2D grid cell indices. If more than one point are projected to the same grid cell in the spatial map, they are accumulated into the cell using an function to aggregate the features or predictions.

**Accumulation.** During *map update* there are many ways to aggregate features or prediction into the map including 1) overwriting the map with the latest observations ($m_t = m_{t-1}$), 2) performing mathematical operations such as max ($m_t = \max(m_t, m_{t-1})$) or mean ($m_t = \text{mean}(m_t, m_{t-1})$), and 3) using a learned neural network. For learned aggregation functions, it is common to use a recurrent network (LSTM, GRU) ($m_t = \text{GRU}(m_t, m_{t-1})$).

During the process of building the map, there are several other important aspects to consider.

**Egocentric vs allocentric.** There are also choices in the reference frame used for map-building, to either maintain a map with an *egocentric* coordinate frame that is relative to the agent (e.g. +y coordinate to the front of the agent) or *allocentric* (e.g. world) coordinate frame.

**Tracking visited areas.** For the map to be complete, it is important for the agent to be able to determine whether it has already visited a location or not, and whether there are unexplored locations. For a specific embodied task, it may not always be necessary for the agent to built a complete map if the task can be accomplished.

**View point selection.** In the case of the embodied setting, the agent is also limited in the possible viewpoints it can observe, and must accumulate into the map, observations in a sequential manner. This is in contrast to the non-embodied setting, where there can be more freedom in selection of viewpoints, and observations can be first collected and then analyzed together.

**Online vs offline map building.** It is possible for the agent to build a map by exploring an environment first. After the map has been built, the agent can then start to perform the specific task. In this scenario, the agent builds the map and performs the task in two separate phases, a process known as *offline* method of map building. Although this method saves compute time during the actual task, there is the overhead of an extra exploration phase for the agent to familiarize itself with a new environment. This approach can be appropriate when the agent is expected to be reused in the same environment repeatedly. However, since the map is a static snapshot and if it is not updated during the task, there can be mismatch between the actual

state of the environment vs what was precomputed. For instance, it might happen that the agent ends up at a location which has not been captured in the map. This might lead to the task failure. Moreover, in real-life applications where a robot is expected to perform a task in an unseen environment, such as search-and-rescue operations, it's not ideal for it to spend extra time exploring the environment first and then performing the task. In contrast a better way is to build or update the map during the task or *online* so as to keep it updated at all times.

**Map building in real world.** Maps built in simulation in embodied AI tasks are often noisy due to unrealistic assumptions that limit its usage in the real-world. Map building has been mostly studied in the community as a sub-module in conjunction to solving more complex high-level reasoning tasks. Researchers have thus tried to investigate what type of maps are useful for which tasks and decoupling the issue of noisy sensors from map building enables them to do exactly that. The most prominent of the assumptions is that of noiseless sensors. For example, sometimes the community assumes *perfect localization* (agent's current location and orientation) at all times during navigation, which is unrealistic in real-world. This is mainly because GPS and Compass sensors are generally noisy, whenever available. However, in most indoor spaces, GPS might not even be available. SLAM methods which work really well in real-world robots operate under the assumption that GPS is not available, and relies on the onboard sensors to estimate its location on the map. Another example of the noiseless sensor assumption is that of a perfect actuation, which means that when an agent initiates an action to move forward by 25cm, it will end up exactly at a location 25cm ahead of its current position. But real-world actuators are noisy and affected by varying friction on different surfaces, which results in significant drifts over time. SLAM systems are inherently capable of addressing such issues by operating under uncertainty in the robot's pose estimation. Loop Closure is a sub-algorithm of SLAM which identifies previously visited locations and then uses them to correct accumulated errors in pose estimation. In general SLAM systems build a more consistent and accurate map of the environment in a real-world setting than the current mapping techniques in embodied AI.

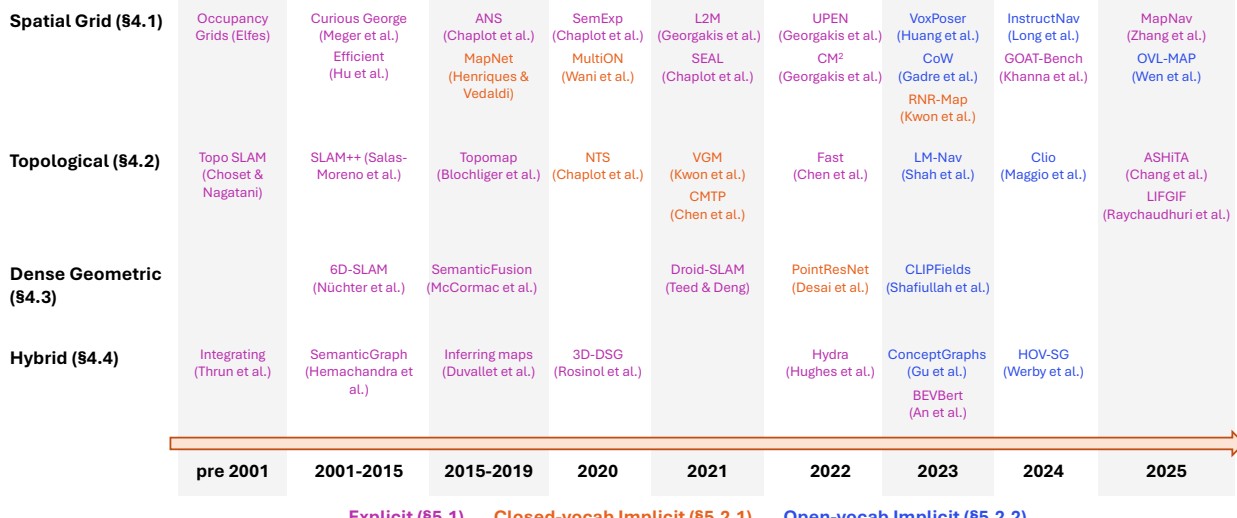

Figure 6: This timeline highlights how semantic mapping methods have progressed over the years, with increasing diversity in map structure and encoding. While various map structures have long been explored, the last few years have seen a shift toward open-vocabulary semantic maps, driven by advances in large language and vision-language models.

# 4 Map structure

In this section we will look at various map structures that have been used in prior works (see Tab. 1) in more depth. A semantic map can be structured in various ways: *spatial grid map*, *topological map*, *dense geometric* or a *hybrid map*. Spatial grid maps are metric maps of the environment such that its dimensions align to that

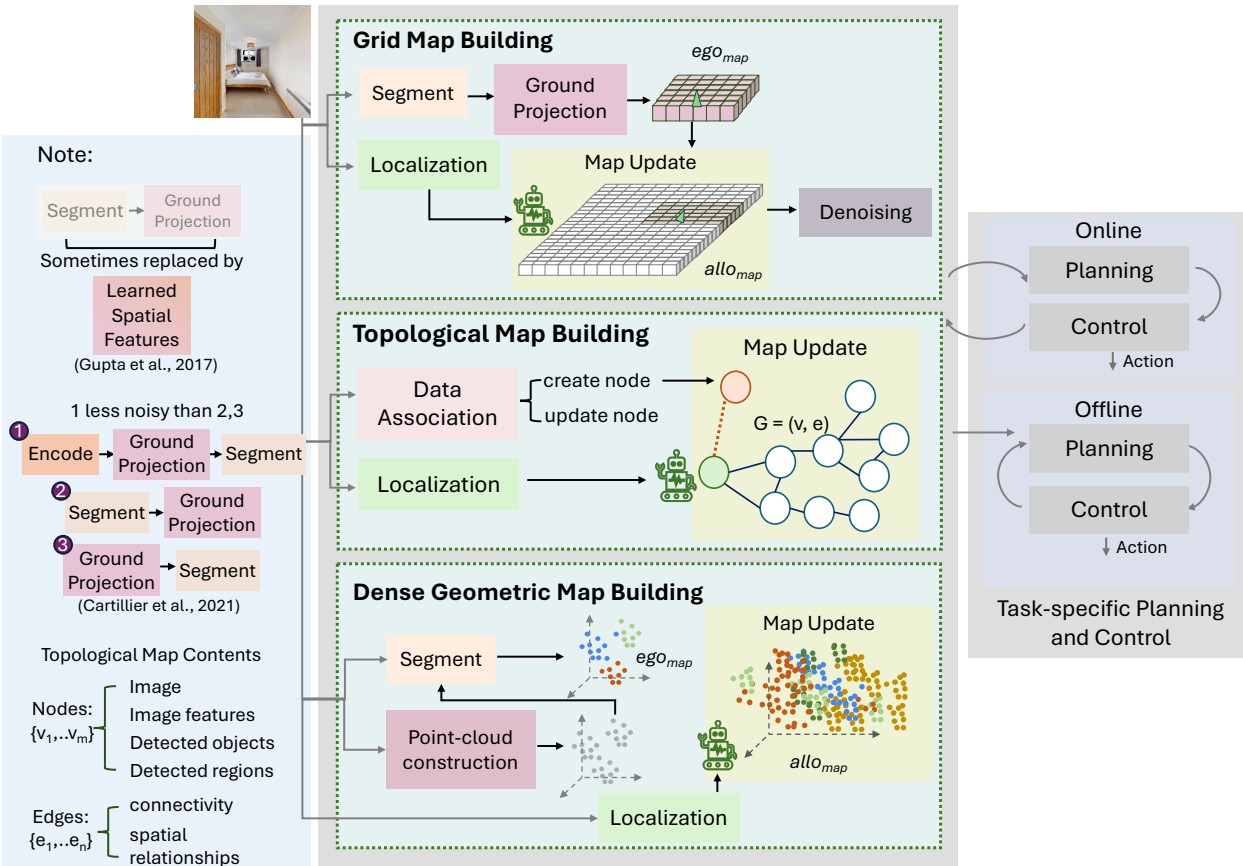

Figure 7: **Grid map building.** A spatial grid map has dimensions $(M \times N \times K)$ where $M$ and $N$ are spatial dimensions and $K$ is the number of semantic channels. The common pipeline to build the map is to *segment* the input image, then *ground project* into an egocentric map $ego_{map}$, which is then registered to the allocentric map $allo_{map}$ via *map update* using the *localized* agent pose. A *denoising* step generally follows and the map is built online along with the task planning and control. While Gupta et al. (2017) learns a spatial representation without segmenting and ground projecting, Cartillier et al. (2021) observes that encoding followed by ground projecting and then segmenting reduces noise in the produced map.

**Topological map building.** A topological map is a graph-like structure, $G = (v, e)$ with nodes ($v = \{v_1, ..., v_m\}$) and edges ($e = \{e_1, ...e_n\}$), that can be built *Online* or *Offline* with or before task-specific planning and control respectively. Based on current observation, the agent performs *localization* on the graph and then performs matching (*Data Association*) with the current node, based on which either a new node is created or an existing node is updated. The nodes and edges may contain various types of semantic information thus enabling decision making in a task.

**Dense geometric map building.** A dense geometric map accumulates semantic information directly onto the 3D geometry of a scene, most popularly using a *point-cloud*.

of the environment and they can be structured as either 2D or 3D grids. Topological maps, on the other hand, represent the environment through a set of landmarks represented as nodes and relation between adjacent landmarks represented as edges in the form of a graph. Dense geometric maps are the densest form of 3D maps whose all three dimensions align to the 3D space such that each 3D point in the scene is captured in the map. Two or more of these types of maps are sometimes combined together to form Hybrid maps.

## 4.1 Spatial grid map

A spatial grid map $m_t$ is a $(M \times N \times K)$ matrix where $M$ and $N$ are the spatial dimensions of the map and $K$ denotes the number of channels to store semantic information at that location. It is a grid like structure where each cell has a width and a height which correspond to a certain area in the physical environment. Current research on indoor embodied AI make use of environments from datasets such as Matterport3D (MP3D) (Chang et al., 2017) and Habitat-Matterport3D (HM3D) (Ramakrishnan et al., 2021) which are 3D reconstructions of real-world spaces. Compared to MP3D, HM3D contains around 10x more scenes with high visual fidelity and lesser reconstruction artifacts. These environments are typically for houses or office spaces with a total area of $< 1000\text{m}^2$. A grid map with each cell representing an area of $400 - 900\text{cm}^2$ is found to be good enough to represent such spaces (Wani et al., 2020; Raychaudhuri et al., 2023). At the start of each episode, the spatial map is initialized with a tensor of size $(M \times N \times K)$, and gradually built as the agent moves around the environment. These are often 2D top-down maps (Gupta et al., 2017; Henriques & Vedaldi, 2018; Narasimhan et al., 2020; Cartillier et al., 2021) with the first two dimensions corresponding to the spatial dimensions of the environment. However, some build 3D spatial maps (Chaplot et al., 2021) to capture the vertical dimension, in which case the map $m_t$ is a 4D tensor $(M \times N \times P \times K)$.

A spatial map may be built in a number of ways depending on whether raw features are directly projected onto the map, or whether a semantic segmentation is used (see Fig. 7). One way is to learn an egocentric projection of the image features as in CMP (Gupta et al., 2017) which forms the egocentric map. In CMP, the egocentric observations are first encoded with a learned image encoder network such as ResNet (He et al., 2016)), and then the network learns to predict an egocentric projection of the image features, without explicit supervision on the map. Instead, the mapper is trained end-to-end along with the planner to predict actions. Egocentric maps, however, not only fail to capture the global structure of the environment, but 'forget' most of the past observations. Thus in long-horizon planning tasks, where the agent needs to 'remember' its past observations for efficiency, egocentric maps fall short.

To maintain an allocentric map, it is necessary to take the egocentric information at each time step and aggregate it into a global map. One way to achieve this is to first obtain egocentric projection of image features and then aggregate to a global allocentric map of the environment via a process known as *registration*. Registration allows the map to incorporate new observations on to specific grid cells. In case the grid cells are already occupied, the new observations are accumulated with the existing ones by employing an aggregation function. This aggregation function can be as simple as taking the latest or the average, but can also be a learned network (see Sec. 3.4). MapNet (Henriques & Vedaldi, 2018) builds an allocentric map by projecting the egocentric image features using depth observations and known camera intrinsics on a 2D top-down grid. This ground projection results in an egocentric projection on a spatial neighborhood around the camera. Next it performs registration by first obtaining a stack of egocentric maps rotated $r$ times and then performing a dense matching with the allocentric map from the previous step to obtain the agent's current pose on the map. The dense matching is efficiently implemented with convolution operators. A LSTM then performs the aggregation of the current observations rotated by the current pose with the allocentric map from the previous step. While an LSTM is used in this work for aggregation, other functions and neural architectures can be used for aggregation as well (see Tab. 3 for a summary of aggregation methods used in different works).

Another way to build an allocentric spatial map is to first convert the image pixels to 3D coordinates in the camera space with known camera intrinsics. The camera coordinates are then converted to world coordinates with known camera pose. Finally the 3D world coordinates are voxelized and projected on a top-down 2D grid with a known projection matrix by summing over the height dimension. This approach is followed in Semantic MapNet (Cartillier et al., 2021) and MOPA (Raychaudhuri et al., 2023). Spatial maps built this way might be noisy due to noisy sensors and need an additional denoising step. Semantic MapNet uses a learned denoising network while MOPA employs a heuristic approach by selecting the centroid of a noisy cluster to obtain a clean map. Following on the last technique, Semantic MapNet (Cartillier et al., 2021) shows that first encoding the image, followed by projecting on to the ground plane and finally performing segmentation on the 2D map reduces noise in the spatial map thereby eliminating the need for an additional denoising step. Irrespective of the approach, it may happen that multiple image features are projected on to the same grid cell. In such cases, it is important to have a scheme for aggregating the features. Some common

approaches to aggregation is to take the maximum (Henriques & Vedaldi, 2018; Wani et al., 2020; Cartillier et al., 2021), mean (Huang et al., 2023a), or the sum of the feature values (Chaplot et al., 2020a).

Table 3: **Spatial grid maps.** Prior works build spatial grid maps (2D or 3D) for various embodied tasks. Information is aggregated onto the map over time in many different ways such as learned recurrent networks and replacing with most recent information among others.

| Method | Task | Environment | Dataset | Aggregation |
|---|---|---|---|---|
| Efficient (2013) | 3D scene analysis | robot | VMR-Oakland (2011), Freiburg (2012) | most recent |
| CMP (2017) | PointNav, ObjectNav | Custom simulator | S3DIS (2016) | weighted mean |
| MapNet (2018) | Mapping | Doom (2016) | Active Vision Dataset (2017) | LSTM |
| ANS (2019) | Exploration | Habitat (2019) | Gibson (2018), MP3D (2017) | channel-wise max-pool |
| MultiON (2020) | MultiON | Habitat (2019) | MP3D (2017) | element-wise max-pool |
| SemExp (2020a) | ObjectNav | Habitat (2019) | Gibson (2018), MP3D (2017) | channel-wise max-pool |
| SemanticMapNet (2021) | ObjectNav, Visual QA | Habitat (2019) | MP3D (2017) | GRU |
| L2M (2021) | ObjectNav | Habitat (2019) | MP3D (2017), Object-Nav (2020b) | most recent |
| SEAL (2021) | ObjectNav | Habitat (2019) | custom with Gibson (2018) scenes | channel-wise max-pool |
| CM$^2$ (2022c) | Vision and Language Nav | Habitat (2019) | VLN-CE (2020) | most recent |
| RNR-Map (2023) | Visual Nav | Habitat (2019) | Gibson (2018) | mean |
| Le-RNR-Map (2023) | Visual Nav | Habitat (2019) | Gibson (2018) | mean |
| VoxPoser (2023b) | Table-top manipulation | SAPIEN (2020), robot | custom | - |
| MOPA (2023) | MultiON | Habitat (2019) | HM3D (2021) | most recent |
| GOAT-Bench (2024) | Multimodal ObjectNav | Habitat (2019) | HM3D-Sem (2023) | most recent |
| Instruction-guided (2025) | Vision-language navigation | Habitat | VLN-CE (2020) | max-pool |

**Summary.** Spatial grid maps capture dense information about the environment. Such representations are useful for the agent to better reason about the spatial structure of the environment. However, the spatial maps need to be initialized with a certain width and height and as such is hard to scale if the environment size changes. Moreover, it consumes a lot of memory which might affect agent performance in the task.

## 4.2 Topological map

Compared to the high-precision grid maps, topological maps are graph-like structures with nodes connected to each other by edges. This essentially abstracts a large space into significant areas (nodes) where the agent can take decisions and connections or paths between them (edges) (Johnson, 2018). This enables parsing the environment into a local and a global structure such that the agent can plan locally in the small space represented as nodes while navigating the large space through graph search following the edges. This way of planning and navigating is inspired from how humans navigate in an unseen environment in that they identify and memorize significant landmarks and find paths to reach those landmarks (Janzen & Van Turennout, 2004; Foo et al., 2005; Chan et al., 2012; Epstein & Vass, 2014). Moreover, in instruction-following tasks, where a robot follows instructions (e.g., 'go to the yellow chair in front of you'), the language instruction is often represented as a graph with high initial uncertainty. As the robot moves and observes its environment, it uses a probabilistic model to update this graph, leading to improved task performance (Tucker et al., 2019; Patki et al., 2019; Arkin et al., 2020a; Raychaudhuri et al., 2025). Thus topological maps have been a popular choice in traditional robotics research such as SLAM (Thrun & Montemerlo, 2006; Rosinol et al., 2020b; Campos et al., 2021), manipulation (Arkin et al., 2020b; Patki et al., 2020), object search (Aydemir et al., 2011; 2013; Lorbach et al., 2014), language grounding (MacMahon et al., 2006; Chen & Mooney, 2011; Tellex et al., 2011a; Matuszek et al., 2013; Duvallet et al., 2013; Gong & Zhang, 2018; Paul et al., 2018) as well as

in embodied AI research (Savinov et al., 2018; Chen et al., 2021; Chaplot et al., 2020b; Kwon et al., 2021; Gu et al., 2024; Mehan et al., 2024; Garg et al., 2024; An et al., 2024; Yang et al., 2024; Tang et al., 2025).

Table 4: **Topological maps.** Various works in indoor embodied AI build topological map either in an exploration phase or online while performing the task. The nodes often store explicit, learned features about the observation or temporal information (visitation timestep), while edges may store relative poses between a pair of nodes or types of edges.

| Method | Task | Map Building Phase | Node values | Node Feature Encoder | Edge values |
|---|---|---|---|---|---|
| Topo SLAM (2001) | SLAM | Online | Pose | ✗ | ✗ |
| SLAM++ (2013) | SLAM | Online | Pose, object type | ✗ | ✗ |
| Imitation (2013) | Instruction following | Online | Pose | ✗ | ✗ |
| TopoMap (2018) | SLAM | Online | free space | ✗ | ✗ |
| SPTM (2018) | ImageNav | Pre-Exploration | Image features | ResNet18 (2016) | ✗ |
| Compact (2019) | Instruction following | Online | Semantic attributes | ✗ | ✗ |
| Multimodal (2020a) | Instruction following | Online | Semantic attributes | ✗ | ✗ |
| NTS (2020b) | ImageNav | Online | Image features | ResNet18 | Relative pose in polar coordinates |
| CMTP (2021) | Instruction following | Pre-Exploration | Image features | ResNet152 | Relative pose in discrete polar coordinates (8 directions, 3 distances (0-2m,2-5m,>5m) |
| VGM (2021) | ImageNav | Online | Image features, visitation timestep | ResNet18 | ✗ |
| Fast (2022) | Path planning | Pre-Exploration | free space | ✗ | ✗ |
| TSGM (2023) | ImageNav | Online | ImageNode stores image features; ObjectNode stores features for detected objects | pretrained image and object encoders | ✗ |
| LM-Nav (2023) | Instruction following | Online | Image features | CLIP (2021) | ✗ |
| RoboHop (2024) | Language querying | Online | Image features for each image segment | CLIP, DINOv2 (2023) | edge types denoting inter- and intra-image connectivity |
| Clio (2024) | Mobile manipulation | Online | object semantics | CLIP | ✗ |
| LIFGIF (2025) | Instruction following | Online | Semantic attributes | ✗ | ✗ |
| ASHiTA (2025) | Task planning | Online | Task, sub-task, objects | MobileCLIP (2024) | ✗ |

A key design decision during topological map building is what should be represented as nodes and what should be edges. Generally speaking, the nodes encode semantic information about locations in the environment such that the agent can make a decision whereas the edges store relationship or connection between the nodes. For indoor navigation, the landmarks for the nodes are typically objects in the environments. They can also be openings or intersections (Fredriksson et al., 2023), locations the agents has visited (Chaplot et al., 2020b), and other regions of interest (Kim et al., 2023; Shah et al., 2023; Garg et al., 2024). For navigation, two nodes are connected with an edge if it is possible to navigate from one node to another. Some methods also store spatial relationships between the nodes (Gu et al., 2024) in the edges to enable better reasoning.

One way to construct a topological map (see Fig. 7) is during an exploration phase previous to the actual task and then use the graph to plan a path to the node most similar to the target, for example, in Semi-Parametric Topological Memory (SPTM) (Savinov et al., 2018). During exploration the agent follows multiple random trajectories for each environment to form a node for every visited location and add an edge between the current node and the previous one to encode connectivity or reachability between them. A common post-processing step includes trimming out redundant nodes and edges to form a sparse graph (Chen et al., 2021). When the graph of one environment is collected from multiple random trajectories, it is also common to merge these graphs into one. However, a topological map generated this way in a pre-exploration phase is still sparse meaning that some observations in the environment might not have been captured by the graph. This affects the agent performance in the downstream task. Moreover, they need a pre-exploration phase which makes them unsuitable for unseen environments.

To mitigate this issue some works construct the topological map online while the agent is navigating during performing the task as is the case with Neural Topological SLAM (NTS) (Chaplot et al., 2020b). NTS consists of several modules – 'Graph Update' to update the topological map from observations, 'Global Policy' to sample subgoals on the map and 'Local Policy' which outputs discrete navigation actions to reach the subgoal. The 'Graph Update' method gradually updates the nodes and edges in the graph from the current observations and agent poses. It first attempts to localize the agent on the graph from the previous timestep. If the agent gets localized in an existing node, it adds an edge between that node and the node from the last timestep. It also stores the relative pose between the two nodes represented as $(r, \theta)$ where $r$ is the relative distance between the nodes and $\theta$ is the relative direction. If the agent is unable to be localized, a new node is added to the graph.

Another important aspect in the topological map creation is how to determine if two observations are similar to each other, in which case the two are mapped to the same node. If they are not similar, two different nodes exist for the two observations. This requires the map building methods to compare RGB images. The goal here is to classify two images as similar if (1) they are exactly the same or (2) there is a slight change in direction or distance between the two. Traditionally this is the problem of *data association* in SLAM-based systems, where an incoming observation could be matched to multiple landmarks (nodes) and either the best match is selected or a new node is created to mark a new landmark (Bowman et al., 2017; Dellaert et al., 2000). In embodied AI some works use a pretrained classifier network to implement this. The network is trained to classify whether two images are from the same area. NTS uses MLP trained with a cross-entropy loss in a supervised manner to predict whether are similar. This however needs annotated pairs of training data. Cross-Modal Transformer Planner (CMTP) (Chen et al., 2021), on the other hand, uses an oracle 'Reachability Estimator' to first obtain the geodesic distance between the two underlying locations based on the traversibility of the 3D mesh. If the distance is below a threshold, it maps them to the same node. Visual Graph Memory (VGM) (Kwon et al., 2021) also uses a pretrained network to determine if two images are similar. But they learn an unsupervised representation of the observations which are then projected onto an embedding space. The idea is to have the embeddings of observations coming from nearby areas clustered together because they are likely to have similar appearances. The training data in this case consist of randomly sampled observations from the training environments, thus eliminating the need for manual annotations. Kim et al. (2023) on the other hand use semantic similarity score obtained from a pretrained network (Li et al., 2021a) between two images to determine whether they are the same nodes. A similar approach is taken in LM-Nav (Shah et al., 2023) where they use CLIP to calculate the cosine similarity between image features. Tab. 4 compares different methods that build topological maps.

**Summary.** In summary, topological maps are convenient to build and maintain due to their concise and condensed representation when compared to spatial maps. They are memory-efficient and can easily be scaled as the environment size increases by simply adding more nodes to the graph. However, they capture only certain landmarks in the environment and as such lack dense global information. This might lead to overlooking visual cues in a cluttered indoor scene that could be helpful for the agent to carry out spatial reasoning.

### 4.3 Dense geometric map

Semantic information can also be directly accumulated onto the 3D geometry of a scene using triangle meshes (Valentin et al., 2013), surfels (Stückler & Behnke, 2014) or point clouds (Jatavallabhula et al., 2023), with point cloud maps being the most widely used due to simplicity and ease of use. While these representations are not always traditionally considered "maps", they effectively function as semantic maps by coupling geometric structure with semantic content. Dense geometric map representations, particularly point clouds, have long been widely used in robotics for tasks such as mapping, localization and navigation, as well as in broader 3D scene understanding tasks (Peng et al., 2023; Xu et al., 2024). Recently, there has been a surge of interest in applying these representations to embodied AI for downstream reasoning and decision-making (Gu et al., 2024). We further categorize dense geometric map representations into *point cloud maps* (Sec. 4.3.1) and *neural fields* (Sec. 4.3.2). Point cloud maps store 3D points with associated semantic labels whereas neural fields represent scenes as continuous functions using neural networks (see Tab. 5 for a summary of prior works).

Table 5: **Dense geometric maps** allow storing semantics onto the 3D geometry of a scene. It can be further categorized into point-cloud maps and neural fields. Here, VL is visual-and-language, Str is Structure, PC is point-cloud, SF is surfel, GS is gaussian splat and NF is neural fields.

| Method | Task | Env | Str | Encoding | Train |
|---|---|---|---|---|---|
| 6D-SLAM (2007) | SLAM | robot | PC | geometry | ✗ |
| Towards semantic maps (2008) | SLAM | robot | PC | 3D location, category label, robot pose | ✓ |
| Robust 3D mapping (2009) | SLAM | robot | PC | geometry | ✗ |
| 3D object-class map (2012) | SLAM | robot | SF | category label | ✓ |
| Parsing (2012) | Semantic segmentation | robot | PC | 3D location, category label | ✗ |
| Street Scenes (2012) | Semantic segmentation | simulator | PC | 3D location, category label | ✗ |
| Dense 3D (2014) | 3D reconstruction | simulator | PC | 3D location, category label | ✓ |
| SemanticFusion (2017) | SLAM | robot | SF | 3D location, normal, category label | ✓ |
| Droid-SLAM (2021) | SLAM | robot | PC | geometry | ✓ |
| Voldor+ SLAM (2021) | SLAM | robot | PC | geometry | ✗ |
| Semantic-NeRF (2021) | Scene understanding | Habitat, Replica | NF | category label | ✓ |
| NeSF (2021) | Semantic segmentation | Kubric | NF | category label | ✓ |
| 3D aware ObjNav (2022a) | Object navigation | Habitat | PC | 3D location, semantic category, consistency information | ✓ |
| ConceptFusion (2023) | Rearrangement, autonomous driving, semantic segmentation | Replica, robot | PC | 3D location, normal, confidence, color, pixel aligned VL features | ✗ |
| OpenScene (2023) | Open-vocab scene understanding | ScanNet, MP3D, nuscenes | PC | CLIP | ✗ |
| LERF (2023) | Open-vocab scene understanding | robot | NF | learned neural VL features | ✓ |
| LGM (2023) | Open-vocab mobile manipulation | robot | GS | CLIP dense patch features | ✓ |
| CLIP-Fields (2023) | Object navigation | Habitat, robot | NF | learned neural VL features | ✓ |
| LangSplat (2024) | Open-vocab semantic segmentation | LERF | GS | VL features from CLIP, SAM | ✓ |
| SemanticGaussians (2024) | Open-vocab scene understanding | ScanNet, LERF | GS | VL features from CLIP, SAM, LSeg | ✓ |
| GaussNav (2025) | Visual navigation | Habitat | GS | category label | ✗ |
| GeFF (2024) | Open-vocab mobile manipulation | robot | GS | MaskCLIP (2022) | ✓ |
| SGS-SLAM (2024a) | SLAM | robot | GS | category label | ✓ |
| GaussianGrasper (2024b) | Open-vocab tabletop manipulation | robot | GS | CLIP | ✓ |
| GeomGS (2025) | Localization | robot | GS | geometric confidence | ✓ |

### 4.3.1 Point cloud maps

Point clouds have become increasingly popular over the years due to their ease of use. Given a 3D point cloud, semantic labels can be associated with each point and thus by coupling geometric structure with semantic content, they serve as a dense semantic map. It is also possible to spatially extend each point to a 3D Gaussian with a covariance matrix. Such representations are called Gaussian splats, and allow for differentiable rendering. Point clouds (and Gaussian splats more recently) have been adopted in a variety of robotics tasks as well as in 3D scene understanding tasks, gaining popularity in the embodied AI community in recent years.

In robotics, point cloud-based mapping has undergone significant evolution, driven by advances in sensing, computational power and SLAM algorithms. Early efforts (Nüchter et al., 2007; May et al., 2009) use 3D sensor scans to build geometrically consistent representations of large-scale environments by emphasizing accuracy and loop-closure. As real-time visual SLAM has matured, dense point cloud generation from RGB-D sensors became feasible, enabling detailed indoor reconstructions (Whelan et al., 2015; Dai et al., 2017; Schops et al., 2019; Min & Dunn, 2021). More recently, learning-based SLAM systems like Droid-SLAM (Teed & Deng, 2021) integrate deep feature extraction and end-to-end optimization to improve SLAM. CoFusion (Rünz & Agapito, 2017) introduce systems for dense SLAM with semantic fusion, pushing toward maps that are not only geometrically accurate but also semantically meaningful. This trend has been accelerated recently with ConceptFusion (Jatavallabhula et al., 2023), which aligns open-vocabulary visual features (e.g., CLIP) with 3D reconstructions to create task-agnostic and queryable maps. These efforts represent a shift from purely geometric maps toward dense, semantically-rich, open-vocabulary representations that support a wide range of downstream tasks. Moreover, Gaussian splatting has recently emerged as a powerful alternative to traditional point cloud representations in robotics, enabling high-fidelity scene reconstruction and differentiable 3D representations useful for tasks such as visual localization, SLAM, and manipulation. Recent works such as SGS-SLAM (Li et al., 2024a) demonstrate how to include semantic labels onto Gaussian splats to effectively perform SLAM, and GaussianGrasper (Zheng et al., 2024b) demonstrates how these representations can be fused with semantics to support open-vocabulary grasping.

In embodied AI, maintaining point cloud maps typically prove to be memory-intensive and not ideal for real-time use. However some efforts have been made to improve efficiency. For instance, Zhang et al. (2023a) builds a 3D semantic scene representation based on an online point cloud-based construction algorithm (Zhang et al., 2020), made efficient by using a tree-based dynamic data structure. This method is very memory efficient and a lot faster when applied to the ObjectNav task. Recent efforts also use Gaussian splatting in navigation tasks. For example, Lei et al. (2025), based on Gaussian splatting, achieves state-of-the-art performance on the ImageNav task.

In 3D scene understanding, point cloud representations have become a cornerstone due to their flexibility and fine spatial resolution. Early learning-based approaches such as PointNet (Qi et al., 2017a) and PointNet++ (Qi et al., 2017b) enable direct processing of unordered point sets for tasks like semantic segmentation and object classification. These were followed by more expressive models like KPConv (Thomas et al., 2019), which introduce learnable convolutions, and Point Transformer (Zhao et al., 2021), which leverage attention mechanisms to capture long-range dependencies across points. More recent work integrates multi-modal and language supervision into point cloud understanding(Peng et al., 2023; Xu et al., 2024), thus enabling fine-grained inference such as semantic segmentation, affordance prediction and 3D object search. OpenScene (Peng et al., 2023), for example, predicts dense 3D features so that they are co-embedded with the corresponding text and the image in the CLIP embedding space. This allows the association of each 3D point in the scene with semantic information such that the scene can be queried using text to infer physical properties, affordances, etc. CLIP2Scene (Chen et al., 2023c) also uses CLIP to perform a 3D point cloud segmentation on outdoor scenes for application in autonomous driving. However, because the optimization happens per scene in many of these methods, they are typically expensive and not suitable for real-time use in embodied applications.

**Summary.** Point cloud maps remain a widely used representation in the research community due to their simplicity, high fidelity and direct correspondence with sensor data. However, challenges persist in handling noise, achieving scalability in large environments and maintaining consistent semantics over time. Future research should aim to integrate multi-modal features and improve robustness to dynamic scenes in point cloud maps, all the while maintaining efficiency.

### 4.3.2 Neural fields

Neural fields are continuous functions that map spatial coordinates (and sometimes viewing directions) to signals such as color, occupancy or semantic features. Unlike point clouds that discretely store information, neural fields encode entire 3D scenes as compact, continuous functions. Often, a neural network (typically MLP) is used to obtain the features associated with each point given the position (x,y,z) as input (Mildenhall et al., 2021; Mescheder et al., 2019). There are typically two strategies to train neural fields 1) use distillation (Kerr et al., 2023; Qiu et al., 2024) to provide features that are similar to a pretrained 2D backbone, such as

CLIP (Radford et al., 2021), and 2) use of differentiable renderer to match the rendered semantics in addition to color (Zhi et al., 2021; Vora et al., 2021).

In robotics, recent works (Sucar et al., 2021; Zhu et al., 2022; Rosinol et al., 2023) have integrated neural fields into SLAM systems for real-time mapping. While these enable compact, high-fidelity 3D reconstruction and view synthesis, they are more computationally intensive and less interpretable than traditional methods. Thus challenges remain in training speed, memory demands and dynamic scene handling. In contrast, neural Signed Distance Fields (SDFs) are a specific type of neural field that encode the signed distance to the closest surface, enabling precise geometric reasoning. Neural SDFs have been shown to be particularly suited for tasks like mapping, collision checking, and planning due to their ability to represent detailed geometry with differentiable distance metrics (Ortiz et al., 2022; Camps et al., 2022; Talha Bukhari et al., 2025). In embodied AI, a representative work using neural field maps include CLIP-Fields (Shafiullah et al., 2023), which learns a continuous, neural implicit representation of 3D environments by aligning 2D image features from CLIP with 3D points in space using volumetric rendering. This enables the agent to build an open-vocabulary, queryable 3D map where each spatial point can be semantically queried using natural language, without needing task-specific supervision.

In 3D scene understanding, neural fields have emerged as a powerful representation from early works on NeRF (Mildenhall et al., 2021) to incorporating semantics into neural fields, enabling tasks such as semantic segmentation, object discovery and scene parsing in 3D (Fu et al., 2022). More recent advances (Wang et al., 2022; Kerr et al., 2023) introduce language-supervised neural fields by integrating CLIP features, enabling open-vocabulary semantic reasoning and text-driven queries over 3D scenes.

**Summary.** Neural fields are gaining popularity in robotics and embodied AI for building continuous 3D scene representations that capture both geometry and semantics. However, challenges remain in real-time training and deployment in dynamic environments. An important future direction would be to focus on making neural fields more efficient, adaptable, and compatible with foundation models for open-world semantic reasoning.

### 4.4 Hybrid maps

So far we have seen how prior works structure maps as either a spatial grid or a landmark based scene graph or a more dense geometric map. However there is a more recent effort on combining these different structures into a single map representation. This helps to capture information at various granularity and perform different types of reasoning on the environment.

The combination of metric information from grid-based maps together with a topological map is also known as a *topometric* map (Thrun et al., 1998b; Tomatis et al., 2001; Blanco et al., 2008b; Konolige et al., 2011; Ko et al., 2013; An et al., 2023; Mozos et al., 2007; Vasudevan & Siegwart, 2008; Zender et al., 2008; Pronobis et al., 2010; Hemachandra et al., 2011; Walter et al., 2013; 2014; Hemachandra et al., 2015; Duvallet et al., 2016; Walter et al., 2022). Thrun et al. (1998b) proposes a single statistical mapping algorithm that first constructs a coarse topological map and uses it to construct a fine-grained grid map. They show that the topological map solves a global alignment problem by correcting large odometry errors, while the grid map solves a local alignment problem by producing high-resolution maps. Tomatis et al. (2001), on the other hand, proposes a compact environmental model where corners and hallways are represented by a topological map and rooms are represented by a grid map, both of which are connected in a single representation. When the robot is moving in hallways, it creates and updates the global topological map, and as soon as it enters a room, it creates a new local metric map[1]. They argue that the robot will only need to be precise inside rooms (e.g. manipulating objects, etc.) which justifies the need for the fine-grained precise metric maps, while the topological map is used to simply maintain global consistency in indistinguishable spaces such as long hallways and transitioning between significant places. BEVBert (An et al., 2023) is a more recent method that constructs hybrid maps offline and then learn a multimodal map representation to perform better spatial reasoning in the complex task of language-guided navigation.

---

[1]They differentiate hallways and rooms by using laser sensor such that thin long open spaces are considered as hallways whereas other open spaces are considered as rooms.

There are also hybrid maps that combines grids, dense geometric, and topological maps. StructNav (Chen et al., 2023b) builds such a hybrid map where the spatial grid stores occupancy information, a scene graph stores landmarks with their connectivity, and a 3D semantic point cloud where each 3D point in the environment has a semantic label associated with it. Thus the spatial grid allows for obstacle avoidance and low-level path planning, the scene graph allows high-level reasoning about the relationship among the landmarks, and the 3D point cloud allows for a more dense semantic and spatial matching.

It is also possible to build maps that capture the semantic hierarchy of scenes at various levels of abstraction (Galindo et al., 2005; Pronobis & Jensfelt, 2012; Hemachandra et al., 2014; Werby et al., 2024) that allows for various levels of reasoning. Armeni et al. (2019) represent a static scene at multiple levels of hierarchy (buildings, rooms and objects), where entities are represented as nodes in a hierarchical graph and connected by edges representing coordinate frame transformations. Tang et al. (2025) (OpenIn) build a similar hierarchical scene graph to track objects in dynamically changing indoor environments. Rosinol et al. (2020a) (Dynamic Scene Graphs) too build layered scene graphs to track moving agents and objects in addition to building a dense 3D metric spatial map by modeling spatio-temporal relations between objects and agents. Hughes et al. (2024) demonstrate that hierarchical scene representations scale better than flat representations in a large environments and thereafter introduces a system called Hydra that incrementally builds 3D scene graphs from sensor data in real-time. Fischer et al. (2024) introduces a multi-level scene graph representation of large-scale dynamic urban environments from a set of images captured from moving vehicles and proposes a new view synthesis benchmark for urban driving scenarios.

**Summary.** Different types of map structures have their own strengths and limitations. While the coarse topological map is useful to represent significant landmarks in an environment, fine-grained metric maps are useful to represent its precise geometry and dense geometric map is useful to represent an even more dense 3D geometry of objects in the scene. Therefore it's crucial to combine two or more of these structures in order to create better representations of the environment, preferably at varying levels of semantic abstraction. While hybrid maps have been explored in robotics, they still remain under-explored in embodied AI. However, weaknesses of these maps need to be considered carefully before combining them. For example, topological and dense geometric maps scale better than a grid map with larger environments, and dense geometric map needs the most and topological map needs the least memory to be stored.

## 5 Map encoding

In this section we will discuss different ways information is encoded and stored into semantic maps. Irrespective of how the map is structured, the map encoding, *i.e.* the values stored in the map can be either *explicit* or *implicit*. An explicit map encoding is one where the type of information stored is clearly known. An implicit encoding, on the other hand, uses a feature embedding that may not be directly interpretable. We now summarize various works that explore these two types of encodings (Fig. 8).

### 5.1 Explicit encoding

Many prior works store explicit information in the map, such as *occupancy*, which has proven beneficial in obstacle avoidance (Elfes, 1989; May et al., 2009; Chaplot et al., 2019; Georgakis et al., 2022a). In such cases, the spatial map stores a binary occupancy value of 1 or 0 depending on whether the corresponding location in the physical environment is 'occupied' by objects or not.

In an exploration task, however, the agent needs to maximize the explored area in an environment while being efficient, in which cases, the information of whether a location has already been explored encourages the agent to explore unexplored areas more. Active Neural SLAM (Chaplot et al., 2019) stores the *explored* (binary) information in addition to the occupancy information. Active Neural SLAM consists of several modules connected together to perform the task of Exploration to maximize the explored area. The 'Neural SLAM' module takes as inputs the visual observations and agent pose and outputs a top-down egocentric spatial map by learning to predict occupancy and explored information using a binary cross-entropy loss. All the modules are jointly trained with different losses. VLMNav (Goetting et al., 2024) similarly stores *explored* information in a top-down voxel map to demonstrate its generalizability to downstream navigation tasks.

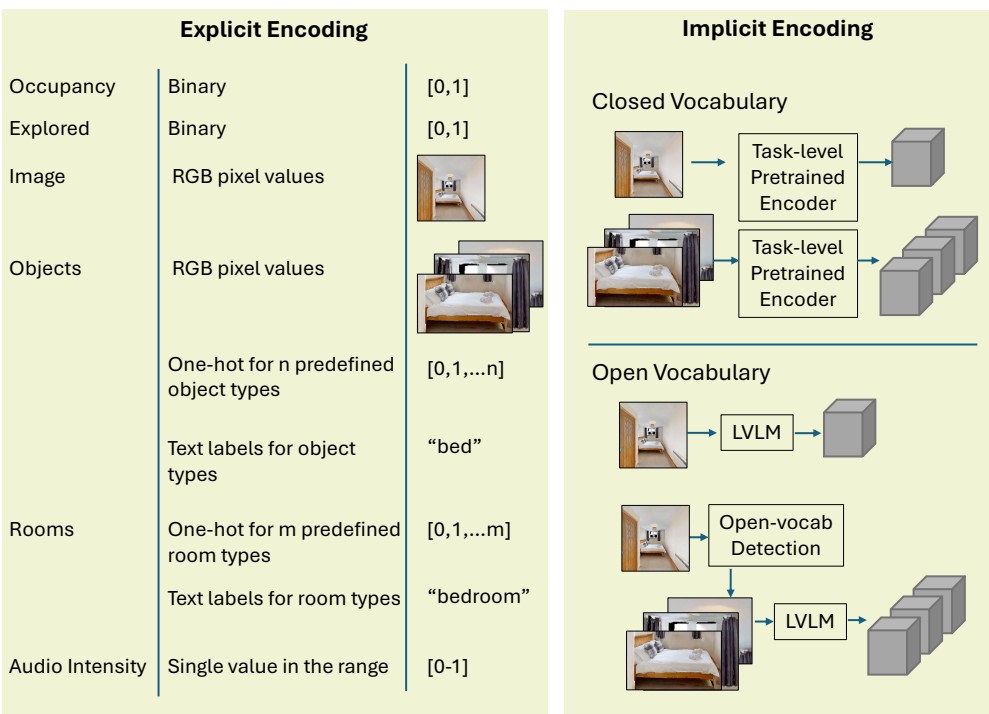

Figure 8: **Map Encoding** refers to the values stored in the map and can be either *explicit* or *implicit*, depending on whether the information is hand-selected or a learned feature representation of the observation.

However, in more complex tasks such as ObjectNav, which requires the agent to navigate to a particular semantic category in its environment, storing occupancy and explored information alone might not be sufficient. In such cases, it is important for the agent to identify the semantic category of the object (Meger et al., 2008; Georgakis et al., 2021; Chaplot et al., 2021; Raychaudhuri et al., 2023; Khanna et al., 2024; Zhang et al., 2025). Goal-oriented Semantic Exploration (SemExp) by Chaplot et al. (2020a) additionally stores the semantic class labels of the objects that the agent identifies through its visual observations. SemExp uses a MaskRCNN (He et al., 2017a) on the RGB observations to predict the semantic categories of the objects and then project these on the map using depth observations. It aggregates the occupancy and explored information using element-wise max pooling whereas the semantic categories are overwritten by the latest prediction. A denoising network is then used to get the final map. This work demonstrates that using this semantic map to predict a long-term goal helps the agent find the goal object category more efficiently. The mapping module is trained using supervised learning with cross-entropy loss on the predicted semantic map. Similarly, a semantic categories map also helps a more complex longer-horizon task of MultiON. MOPA (Raychaudhuri et al., 2023) shows that maintaining a memory with objects that the agent observes while moving around is crucial to perform this task efficiently. However, a map built by segmenting the image first and then projecting may result in 'label splattering' *i.e.* noisy category labels splattered across multiple grid cells in a spatial map. This arises mostly due to noisy depth observations and might negatively affect agent performance in a task. Semantic MapNet (Cartillier et al., 2021) finds that projecting encoded features on to a map and then segmenting produces a more noise-free map. They show that this map can be then be applied to two different tasks effectively. However this method requires an additional exploration phase where the agent effectively explores the environment to build the map first and then use that map in the downstream task. A recent work GOAT (Chang et al., 2023; Khanna et al., 2024) shows that having an object instance map helps to navigate to a goal specified by either language, image or a category label. They achieve this by storing raw images and later using CLIP (Khandelwal et al., 2022) for image-to-language matching and SuperGlue (Sarlin et al., 2020) for image-to-image matching. MapNav (Zhang et al., 2025) builds an annotated semantic map by storing text labels of the segmented objects and shows that this helps a VLM to ground objects better.

While a map with semantic categories help in object navigation, an acoustic map storing audio intensity is found to be useful in the Audio-Visual Navigation task (Chen et al., 2020b). Here the map is aggregated by averaging the intensity. In the more complex Interactive Question Answering task, Gordon et al. (2018) find that storing object detection probabilities in a spatial map helps agent performance. They use a GRU recurrent memory to aggregate the current map with the previous one.

While the above works build explicit spatial maps, some works also build explicit topological maps (Choset & Nagatani, 2001; Blochliger et al., 2018; Chen et al., 2022; Salas-Moreno et al., 2013; Duvallet et al., 2013; Patki et al., 2019; Arkin et al., 2020a; Raychaudhuri et al., 2025; Chang et al., 2025). Kwon et al. (2021) store the visitation timestep in the graph node in order to encode a temporal relation between visited locations. This information is then replaced by the latest visitation timestep while aggregating the map.

**Summary.** The advantage of explicit map encoding is its interpretability and the fact that it allows investigating the type of information that is beneficial for various downstream tasks. However, the type of semantic information to be stored is a design choice based on the task. Moreover, the above approaches require a predefined set of categories to be mentioned beforehand to the mapper. This restricts the map to store only a limited number of object categories.

## 5.2 Implicit encoding

Implicit maps store latent features in a semantic map. While most of the prior works use extracted features from a vision model pre-trained on a closed-vocabulary set of object categories, recent methods use features extracted from a pre-trained large vision-language model thus producing flexible open-vocabulary queryable maps. It is also possible to store features that are not necessarily queryable with language, but captures the visual information at that location. One example is RNR-Map (Kwon et al., 2023) which uses a grid map with latent codes that corresponds to a neural field that can be used to render possible views at that location.

### 5.2.1 Closed-vocabulary encoding

These features can be learned from scratch during training. For example, (Wani et al., 2020) learn image features using CNN blocks and use them to build a global spatial map of the environment. This is trained end-to-end to predict actions in the MultiON task. On the other hand, the features may also be extracted from a vision model such as ResNet, pre-trained on the ImageNet (Deng et al., 2009) data to encode RGB images. For example, (Gupta et al., 2017) introduces Cognitive Mapper and Planner (CMP) that uses a pretrained ResNet-50 model to encode the egocentric RGB images and then projecting on the map using a differentiable mapper module. In CMP, the learned map encoding is not explicitly supervised but learned in conjunction with a differentiable planner. This enables the mapper to learn to store information that is most useful for the planner to perform the task efficiently. Also the map is accumulated over time meaning that the map from one navigation step is integrated into the next using a differentiable warp. Similarly MapNet (Henriques & Vedaldi, 2018) also uses a pretrained ResNet-50 model to extract image features but they build an allocentric global map of the environment instead of an egocentric map. They do so by first ground projecting the features and then registering these into a global allocentric map, updating the map at every navigation step. This model is learned end-to-end on the task of localization and trained with a series of RGB-D observations and the corresponding ground-truth positions and orientations.

While the above methods build a spatial map, other methods use a similar approach to store implicit features in the nodes of a topological map. Each node stores encoded features from the observations at a particular location in the environment. Here too, using a pretrained ResNet encoder to extract RGB image features is a popular choice among prior works (Chaplot et al., 2020b; Chen et al., 2021).

**Summary.** The advantage of using a pretrained ResNet model over learning from scratch is that it has already been trained to encode useful features and is thus sample efficient. However, using a pre-trained ResNet is limited by the number of object categories it was trained on.

Table 6: **Open-vocabulary maps.** Various works build open-vocabulary semantic map using trained as well as off-the-shelf pretrained models in both simulation and real-world robots. These methods use either heuristics-based or LLM-based planner to perform the downstream taskss.

| Method | Environment | Training | VL Encoder | Task Planner | Aggregation |
|---|---|---|---|---|---|
| CoW (2023) | Habitat, RoboTHOR | ✓ | CLIP | A* | highest similarity score |
| VLMap (2023a) | Habitat, robot | ✗ | LSeg | A* | mean |
| NLMap (2023a) | robot | ✗ | CLIP, ViLD | LLM-based | multi-view fusion |
| ConceptGraphs (2024) | AI2Thor, robot | ✗ | CLIP, DINO | GPT-4 | highest similarity score |
| VoxPoser (2023b) | Sapien, robot | ✗ | OWL-ViT | GPT-4 | - |
| VLFM (2023) | Habitat, robot | ✗ | BLIP-2 | pretrained PointNav | highest similarity score |
| CLIP-Fields (2023) | Habitat, robot | ✓ | CLIP | SLAM | weighted mean |
| LM-Nav (2023) | robot | ✗ | CLIP | modified Dijkstra | CLIP similarity |
| ConceptFusion (2023) | AI2Thor, robot | ✗ | CLIP | LLM-based | weighted mean |
| Le-RNR-Map (2023) | Habitat | ✓ | CLIP | Random walk exploration | latest |
| HOV-SG (2024) | Habitat, robot | ✗ | CLIP | A* | weighted mean |
| RoboHop (2024) | Habitat, robot | ✗ | CLIP | Dijkstra | mean |
| Clio (2024) | robot | ✗ | CLIP | Dijkstra | mean |
| OneMap (2025) | Habitat, robot | ✗ | SED | A* | weighted sum with uncertainty-based weights |
| OVL-Map (2025) | Habitat | ✓ | LSeg | DD-PPO (2019b) | weighted mean |

### 5.2.2 Open-vocabulary encoding

The limitation of the closed-vocabulary encoding can be mitigated by extracting features from a Large Vision-Language Model (LVLM) that was jointly trained on a vast amount of internet data of images and their text captions, such as CLIP (Radford et al., 2021). This allows the map to store information about 'any' object in the environment and eventually be queried via an open-vocabulary text query (Wen et al., 2025) not limited to a predefined set of object categories. For example, an agent may be asked to 'find a *red and blue striped zebra toy* in the children's room'. It is likely that the agent has not seen a 'red and blue striped zebra toy' during training but can leverage a LVLM to reason about its prior knowledge about zebras, colors and rooms in general. Moreover, recently large language models (LLMs), such as GPT-4 (OpenAI, 2023), have been shown to be able to perform complex task planning. Thus open-vocabulary maps built using LVLM along with LLM-based planners (Taioli et al., 2023; Huang et al., 2023b; Long et al., 2024) have led to a recent line of works in embodied agents (Tab. 6, Fig. 9).

A pretrained CLIP model can be used to compute similarity scores between an input image and a natural language description with the highest score corresponding to the most likely image-text match. CLIP has been successfully used in map building where the map stores the similarity scores between each image that the agent observes and the language instruction describing an object. Popularly, these maps are called *value maps*. CoW (Gadre et al., 2023) shows that such a 2D value map can be successfully applied in the downstream task of language-driven ObjectNav in a zero-shot manner without any re-training. The planner in this method plans a path to the object when the stored similarity score exceeds a certain threshold. VLFM (Yokoyama et al., 2023) follows a similar strategy to perform ObjectNav task by using the BLIP-2 (Li et al., 2023) 2D value map to semantically explore the environment. InstructNav (Long et al., 2024) extends this idea to enhance semantic value maps with multi-sourced value maps encoding actions, landmarks, and navigation history, thus improving generic instruction following. VoxPoser (Huang et al., 2023b), on the other hand, builds a similar 3D value map to efficiently perform table-top robot manipulation. Such map encoding is quite

powerful compared to the previous encodings that stores the semantic labels for a predefined set of objects. However, this particular map still lacks the ability to perform spatial reasoning because it performs similarity matching to the entire input image ignoring the semantic information about individual objects in the scene.

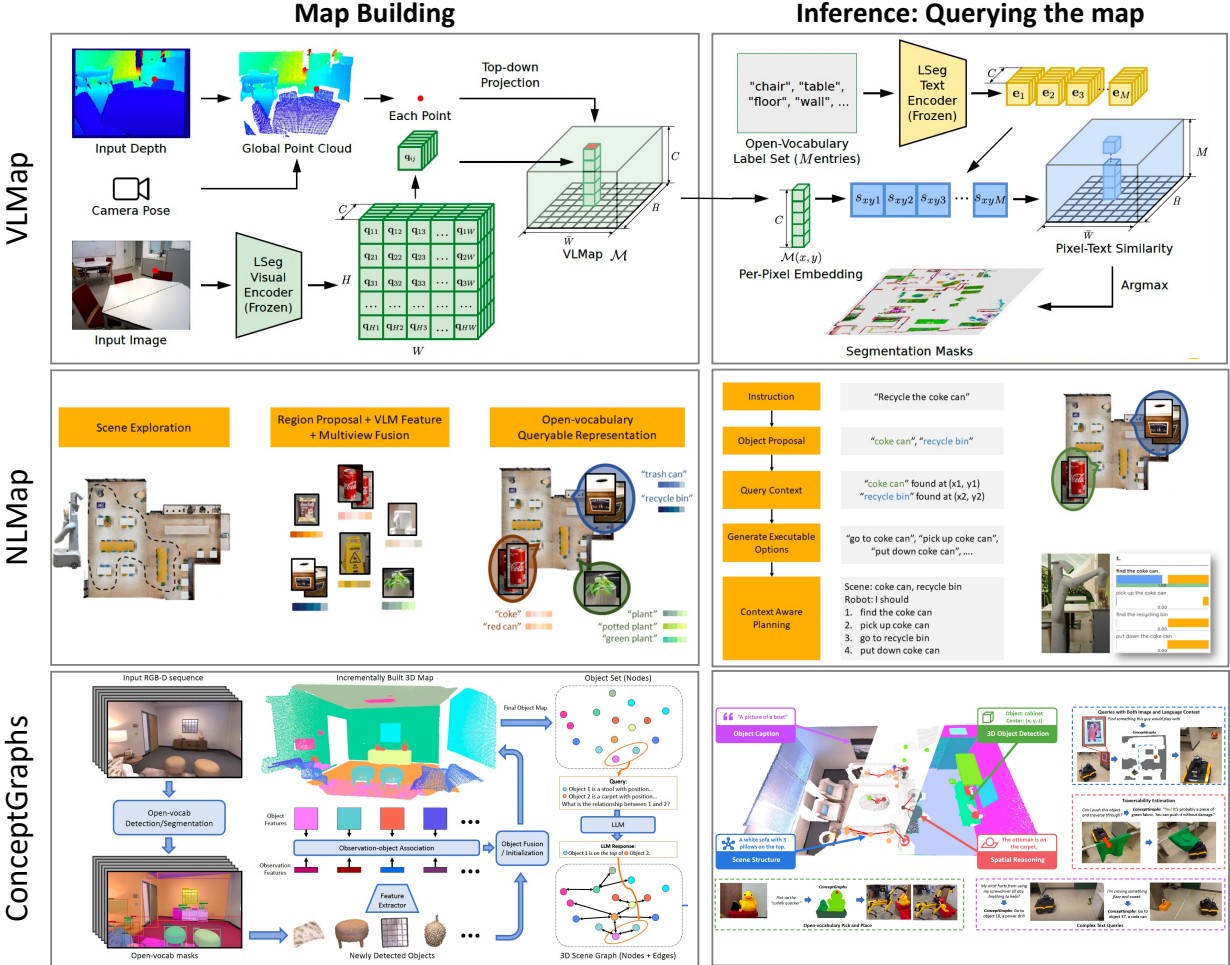

Figure 9: **Open-Vocabulary map building.** There has been a growing interest to build flexible open-vocabulary maps which can be built once and then used in various downstream tasks during inference. VLMap (Huang et al., 2023a) and NLMap (Chen et al., 2023a) structure their maps as spatial maps while ConceptGraphs (Gu et al., 2024) build a topological map. *(Figures reproduced from paper.)*

To mitigate this issue, researchers need to develop methods that identify where objects are located in the image, and then extract features for those objects. One way to achieve this is to store feature embeddings for each pixel in the input image. This can be done using LSeg (Li et al., 2022) which outputs pixel-level embeddings given an image as is done in VLMaps (Huang et al., 2023a). The embeddings are then projected into a 2D spatial map using depth observations. When multiple points are projected onto the same grid cell on the map, VLMaps aggregates by averaging the features. During inference, they extract object names from the language query and calculate pixel-text similarity scores on the map to retrieve the objects of interest. However, these pixel-level embeddings are very dense and can be redundant since not all pixels in an image contain objects. Second, some information might be lost while averaging the pixel-level embeddings. A third issue is that it does not encode object-level semantics, thus ignoring information about spatial relationships. OneMap (Busch et al., 2025) stores patch-level features extracted from SED (Xie et al., 2024) with a hierarchical encoder-based backbone which has shown to capture spatial information better than the transformer-based architecture. This alleviates the issues of using pixel-based features.

However, a better way is to first identify all objects present in an image, which can be done by using a class-agnostic region proposal network to propose regions of interest (objects) in an image. This approach is used in NLMap (Chen et al., 2023a) which uses ViLD (Gu et al., 2021) as its class agnostic region proposal network. For each region of interest, they extract image embeddings using an ensemble of CLIP and ViLD. These features are then stored in a 3D spatial map along with the 3D location and estimated size of the object. They show that this spatial map representation can be applied to any downstream task by performing a natural language based query. This is achieved by extracting object names from the query and using them to select the object from the map with the highest similarity scores.

While NLMap builds a 3D spatial map, many (Shah et al., 2023; Garg et al., 2024) build a topological map following a similar approach. For instance, ConceptGraphs (Gu et al., 2024) is a recent approach that retrieves the objects of interest from the input image by using the class-agnostic 2D segmentation model, Segment-Anything (SAM) (Kirillov et al., 2023) to obtain candidate masks. These objects then form the nodes of the topological map. The image features for each object can then be extracted by CLIP and DINO (Oquab et al., 2023) to be stored in the corresponding node. Additional information about the objects can also be stored in each node. For example, ConceptGraphs store the point cloud of the proposed masks and a caption for each object as obtained by using LLaVA (Liu et al., 2023a) and GPT-4 (OpenAI, 2023) along with a point cloud obtained by projecting the object mask proposed by SAM into 3D space. After the nodes are formed, edges between the nodes can be constructed depending on whether the objects are spatially related. In ConceptGraphs, spatial relation between two objects is determined by whether the point clouds of the respective objects have a geometric similarity or overlap. In other words, when a certain proportion of points in point cloud of one object lie within a distance threshold of that of the second object, the objects can be said to be spatially related to each other and an edge is constructed between the corresponding nodes. The edges additionally store a spatial relationship description obtained from LLM. When a newly detected object is found to be similar to a node, it is updated with averaged object features, union of point clouds and the latest object caption. In case it is not similar, a new node is added to the graph. ConceptGraphs show that a single map representation built in this manner can be successfully and effectively used in object grounding, robot navigation and robot manipulation tasks.

**Summary.** The advantage of an open-vocabulary map encoding is that it can be built once and then transferred to several different downstream tasks. It can be queried using an open-vocabulary text effectively and is highly interpretable. However one current limitation is that the computational costs of using large foundation models can be significant.

## 6 Evaluation

In the embodied AI and robotics literature, very little focus has been given to evaluating the built map (*intrinsic evaluation*). Instead the focus has been to evaluate the agent performance on the downstream tasks (*extrinsic evaluation*) through various metrics. This is because these works build semantic maps as an intermediate step while performing downstream tasks, such as localization, navigation, exploration, manipulation and so on. Moreover, most of these works focus on a single task and hence it suffices for them to evaluate the task performance directly without caring about how well the map representation is. However, we think that it is crucial to do intrinsic evaluation based on the *accuracy*, *completeness*, *consistency* and *robustness* of the built map, in addition to evaluating the task performance. We next discuss the evaluation metrics (both extrinsic and intrinsic) in detail.

### 6.1 Extrinsic evaluation

This section discusses extrinsic (or task-level) evaluation metrics commonly used to assess agent performance across embodied tasks. In navigation, success is not simply reaching the goal but signaling it intentionally, often via a dedicated action like 'Stop' or 'Found' (Anderson et al., 2018a). Accordingly, Success Rate (*SR*) is used to evaluate whether the agent correctly signals goal completion, while Success weighted by inverse Path Length (*SPL*) accounts for trajectory efficiency. These metrics are standard in navigation benchmarks (Beeching et al., 2020; Batra et al., 2020b; Chaplot et al., 2020a;b; Wijmans et al., 2019b; Li et al., 2021c; Krantz et al., 2022; Georgakis et al., 2022a; Taioli et al., 2024; Yokoyama et al., 2024). Additional metrics include

Oracle SR (*OSR*), which accounts for when the agent reaches the goal but fails to signal, and Navigation Error (*NE*), which measures the final distance to the goal. In multi-step tasks or multi-goal navigation, metrics like *Progress* (Wani et al., 2020; Khanna et al., 2024) or *Goal-Condition Success* (Shridhar et al., 2020; Padmakumar et al., 2022) capture the proportion of successfully completed subgoals. For instruction following, normalized Dynamic Time Warping (*nDTW*) is used to evaluate how well the agent's trajectory aligns with the reference path (Ilharco et al., 2019; Krantz et al., 2020; Raychaudhuri et al., 2021; 2025). Tasks involving classification or detection often use *precision* (how often predictions are correct), *recall* (how well all relevant targets are found), *F1 score* (combines and balances precision and recall) and receiver-operating characteristic curve or *ROC* curve (measures the tradeoff between sensitivity or true positive rate and specificity or false positive rate) (Chen & Mooney, 2011; Magnusson et al., 2009; Matuszek et al., 2012; Tellex et al., 2011b; Wellhausen et al., 2020) to evaluate prediction quality. In exploration tasks, *coverage* is used to measure the proportion of the environment observed (Chaplot et al., 2019). In manipulation tasks such as object rearrangement or tabletop operations, *task success rate*, measuring the percentage of trials where the agent achieves the desired manipulation (e.g., grasping, placing), is the dominant metric (Kalashnikov et al., 2018; Batra et al., 2020a; Lin et al., 2020; Zeng et al., 2020; Huang et al., 2023b; Gu et al., 2023). Some works also evaluate policy outputs using precision (Zeng et al., 2017), or employ pose error metrics to assess deviations between the final and target object positions (Liu et al., 2012).

**Summary.** Extrinsic evaluation metrics have been extensively studied across the literature (Deitke et al., 2022) and are continually evolving in response to the growing complexity and diversity of embodied tasks. As new challenges emerge in navigation, manipulation, and instruction following, the need for nuanced and task-specific performance metrics has driven the development of more robust and comprehensive evaluation frameworks. Although extrinsic metrics have dominated evaluation in the field, we believe it's time for the community to place greater emphasis on intrinsic metrics (we discuss this in next section). This shift is especially important given the recent progress in building open-vocabulary, flexible, and general-purpose maps, which require more nuanced assessment of map quality beyond task success alone.

## 6.2 Intrinsic evaluation

In this section, we discuss intrinsic evaluation or map-level evaluation which evaluates the map directly. While most robotic and embodied AI systems are evaluated based on downstream task success, a few focus on directly evaluating the quality of the maps built during these tasks. However, there is no widely agreed-upon metric suite for evaluating the map representation itself. We next list the various dimensions across which we believe the map can be evaluated and discuss the gaps and future scope for each.

**Accuracy.** Map accuracy refers to how accurately the map captures geometric or semantic information when compared against the ground truth. However obtaining the ground truth map can be challenging in most cases. Some works (Georgakis et al., 2022a; Ramakrishnan et al., 2020) evaluate the 2D occupancy maps by using *map accuracy* to measure how much of that area matches with the ground-truth, and *IoU* to measure intersection over union between the built and the ground-truth map. While others (Cartillier et al., 2021; Georgakis et al., 2022b) evaluate the accuracy of the built 2D semantic map using semantic segmentation metrics such as *pixel-wise labeling accuracy*, *pixel-based F1 score*, *IoU* score and a *contour-based average boundary F1 score*. However, there is a lack of standardized metrics that can be used across various map types. For example, IoU is not directly applicable to topological maps. Moreover, while evaluating accuracy for semantic maps that store a fixed set of object categories is straightforward, it is tricky to evaluate semantic maps that store open-vocabulary features. As an alternative, OpenScene (Peng et al., 2023) performs semantic segmentation on their open-vocabulary map for a fixed set of categories and thereafter evaluate it using accuracy and IoU. ConceptGraphs (Gu et al., 2024), on the other hand, evaluate their open-vocabulary topological map by employing human evaluators on Amazon Mechanical Turk (AMT) to report the scene graph accuracy. OpenLex3D (Kassab et al., 2025) has been recently introduced to evaluate open-vocabulary 3D scene representations, by providing object labels at various granularity or precision. Similar evaluation techniques could benefit intrinsic evaluation in embodied setting as well. The challenge to evaluate modern-day open-vocabulary maps lies in collecting precise ground-truth maps for large real-world environments, which is labor-intensive or infeasible. This limits the use of direct comparisons in order to

calculate map accuracy. Moreover, accuracy in dynamic or cluttered environments is also rarely assessed. Both these provide possible future area of research.

**Completeness.** Map completeness measures how completely a generated map represents an environment, encompassing both geometric and semantic coverage. Geometric coverage refers to the fraction of environment that has been mapped whereas semantic coverage refers to the completeness of the semantic information captured in the map. Completeness is particularly critical for tasks like search and rescue, where a thorough understanding of the surroundings is essential. Moreover, complete geometric and semantic mapping support better decision-making by reducing dependence on incomplete or inaccurate data. The extent of map completeness depends on how thoroughly the robot explores the environment during the downstream task and is closely tied to the 'stopping criteria' – a method that determines when to end the exploration process. In embodied AI, exploration typically ends when the task is deemed complete or when a predefined time budget is reached. However, reaching this time budget does not always guarantee that the map is fully complete, making the development of reliable stopping criteria an ongoing challenge in robotics as well as embodied AI research (Placed & Castellanos, 2022; Luperto et al., 2024). Geometric coverage metric has been reported in prior works that performs the task of exploration (Chaplot et al., 2019) in terms of the fraction of the environment explored by the robot. However, a lot of works that tackle other embodied AI tasks fail to report this metric even if exploration is a crucial part of their task (Gervet et al., 2023; Raychaudhuri et al., 2023; Yokoyama et al., 2023). Semantic coverage on the other hand relies on the presence of a detailed ground-truth map with semantic information and could be hard to obtain, as discussed in the previous paragraph 'Accuracy'. Due to the lack of detailed ground-truth semantic maps, semantic completeness is rarely quantified.

**Consistency.** *Geometric consistency* of a map refers to how accurately the spatial structure (distances, angles, and relative positions of structures/objects) of the map represents the physical layout of the environment. Accurate geometry ensures safe and efficient path planning and obstacle avoidance. This is crucial in classical robotics for identifying loop closure, a popular technique to recognize previously visited locations in SLAM, which refines the map and reduces drift. Popular loop closure metrics (Li et al., 2021b; Qian et al., 2022) include Absolute Trajectory Error (*ATE*) that evaluates the cumulative error, in addition to precision and recall. However, in embodied AI systems, there is no drift due to absence of sensory and actuator noise and hence the generated maps are mostly consistent with the environment. Hence prior embodied AI works do not report geometric consistency metrics for the built map. That said, ensuring geometric consistency will be crucial in future where dynamic moving objects may disrupt the map's structural fidelity. It can be measured by reporting the Root Mean Square Error (*RMSE*) from the ground-truth geometry. On the other hand, the *semantic consistency* of a map refers to the alignment between the semantic information of structures/objects and their physical locations in the environment. Semantic consistency is particularly crucial to remain consistent over time despite changes in perspective, lighting, or environment dynamics when the robot moves around the environment. This could be measured using a temporal accuracy metric that measures how accuracy of the semantic information changes over time. However, none of the prior works measure semantic consistency and may be considered in future research.

**Robustness.** Evaluating robustness in semantic maps is mostly crucial for assessing their reliability in unpredictable or dynamic environments. A robust map exhibits low uncertainty and high confidence in its semantic information, allowing the robot to adapt to errors, sensor noise, and environmental changes. In SLAM, some study system robustness to measure Absolute Trajectory Error (*ATE*) and Relative Pose Error (*RPE*) under noise perturbations (Prokhorov et al., 2019; Yang et al., 2025). On the other hand, since recent approaches use pretrained models, measuring the confidence of the model's predictions could be beneficial to assess map robustness, with higher confidence indicating a more robust map. Alternatively, model uncertainty can be measured by assessing the variance in model predictions, reflecting epistemic uncertainty in the model, where low uncertainty will indicate a robust map. While some studies in embodied AI (Georgakis et al., 2022a; Raychaudhuri et al., 2025) incorporate uncertainty into task planning, it has not been used as a formal metric in previous research, presenting an area for future exploration. We think that uncertainty-based reasoning in the SLAM and active robotic exploration can provide valuable insights into designing uncertainty-based map robustness metrics. More specifically, *joint entropy*, *expected map information*, and *uncertainty-aware mapping* all relate to reasoning under uncertainty in the robotics literature. Joint entropy quantifies the total uncertainty over both the robot's trajectory and the map, while expected map information (often expressed

as mutual information) captures how much that uncertainty in the map is expected to decrease after a future observation. These are widely used as internal utilities for planning (Stachniss et al., 2005; Blanco et al., 2008a; Georgakis et al., 2022a; Carlone et al., 2013), guiding action selection in active SLAM and exploration by prioritizing trajectories that maximize expected information gain. Uncertainty-aware mapping refers to the practice of explicitly representing and propagating uncertainty within the map itself, often using probabilistic models like occupancy grids with confidence estimates, Gaussian processes, or learned uncertainty models. This enables more robust perception and safer navigation, especially in dynamic or partially observed environments. While joint entropy and expected map information are used to decide where to go next, uncertainty-aware mapping focuses on how we represent what we've already seen. However, these are rarely reported as evaluation metrics, as they are task-dependent, abstract, and computationally intensive to calculate precisely.

Table 7: **Map evaluation.** While extrinsic or task-level evaluation has been extensively studied over the years, there has been little progress in intrinsic evaluation to assess the quality of the built map across various dimensions such as accuracy, completeness, consistency and robustness. In this study, we emphasize the need for a standardized suite of intrinsic metrics to be applied across different map structures and encodings.

| Evaluation | Dimensions | Metrics | Task |
|---|---|---|---|
| Extrinsic | Utility | Success rate, SPL, NE, OSR
nDTW
QA accuracy, Nav accuracy
Precision, Recall, F1, ROC
RPE (translation, rotation) | Navigation (2020; 2020b; 2019b; 2021c; 2024; 2024)
Instruction following (2020; 2025)
Embodied QA (2019a; 2024)
Language grounding (2011; 2012; 2011b)
Localization (2019) |
| Intrinsic | Accuracy
Completeness
Consistency
Robustness | Map accuracy, IoU, Pixel F1, Boundary F1
Geometric coverage
Precision, Recall, True Positives, ATE
ATE and RPE under noise perturbations | Semantic segmentation (2021; 2021)
Exploration (2019)
Loop closure (2021b; 2022)
Visual SLAM (2019) |

## 6.3 Summary

In this section we summarize the key challenges and the future scope of evaluation (Tab. 7). Extrinsic evaluation metrics have been widely explored in the literature and continue to evolve with the emergence of increasingly complex tasks. However, evaluating the quality of maps (intrinsic) in semantic SLAM and embodied AI continues to be understudied and presents two key challenges. First, collecting accurate, fine-grained, open-vocabulary ground-truth annotations for large, real-world environments is inherently difficult. This limitation makes it particularly challenging to assess important properties of the built maps such as semantic accuracy, consistency, and completeness of the maps. Without reliable ground truth, these evaluations often remain coarse or dataset-specific. Second, there is currently no standardized suite of evaluation metrics that can be consistently applied across different types of map structures (topological, spatial grid, dense geometric, etc.) or across different encoding schemes (explicit and implicit features). Together, these challenges highlight a critical gap in the field and point toward the need for developing scalable, generalizable map evaluation frameworks, offering an important direction for future research.

## 7 Challenges

Despite significant progress in semantic map building, numerous challenges remain that warrant further attention and improvement. This section outlines key open problems in the field across multiple dimensions.

**Efficiency.** As semantic maps grow richer by incorporating fine-grained semantic information, they become increasingly data-heavy, posing significant challenges for memory efficiency and compact storage, particularly on robots with constrained hardware. The underlying map structure plays a crucial role in determining storage and computational efficiency. Different representations offer trade-offs among semantic richness, spatial fidelity, and scalability. For example, spatial grids uniformly capture space but are memory-intensive and difficult to scale in large, high-resolution environments. Dense geometric maps provide high spatial fidelity and support per-point semantic features (e.g., color, normals), yet are often redundant and storage-heavy. In

contrast, topological maps are lightweight and scalable but lack the geometric detail needed for tasks like manipulation or precise localization. Hybrid topometric maps attempt to balance these aspects by combining the spatial accuracy of grids with the efficiency of topological graphs, though managing and updating both layers increases complexity and computational load. Map encoding methods also impact efficiency. Learned high-dimensional representations (e.g., neural features) offer compressed yet expressive encodings, but they are often harder to interpret or update, and computationally expensive to query. Overall, achieving efficient storage and memory usage in semantic mapping remains an open and active research challenge.

**Scalability.** As semantic mapping systems are deployed in increasingly large, dynamic, and diverse environments, scalability emerges as a critical challenge. It refers to a system's ability to accommodate growth in spatial extent (e.g., large physical spaces), semantic complexity (e.g., diverse object categories and scene types), and temporal evolution (e.g., long-term and lifelong operation), while maintaining efficiency, accuracy, and robustness. High-resolution, globally consistent maps require significant computational and memory resources, which grow rapidly with the size and richness of the environment. For example, a home robot navigating multiple cluttered rooms over extended periods must retain detailed spatial layouts and object-level information without exhausting onboard storage or processing capacity. The challenge intensifies in dynamic settings, such as when furniture is rearranged or objects are moved, requiring the map to adapt continuously. Moreover, as maps expand, core operations like loop closure, re-localization and map querying become increasingly expensive in both simulation and real robots. This underscores the need for lightweight, adaptable semantic mapping methods that can scale with environmental complexity while staying efficient on resource-constrained hardware.

**Real-time processing.** Real-time processing is a critical challenge in semantic map building, especially in applications such as autonomous driving and real-time human-robot interaction, where decisions must be made within strict latency bounds. Semantic mapping in these domains involves constructing maps with semantic and spatial integrity at frame rates fast enough to support safe and responsive behavior. However, this is extremely challenging. For instance, an autonomous vehicle navigating an urban environment must simultaneously detect traffic signals, update the map with moving vehicles or pedestrians, and plan a safe trajectory in milliseconds. Any lag in perception or map update could lead to disastrous decisions. Balancing the trade-off between semantic richness and processing speed, often on memory-constrained hardware, remains an open research challenge. Advances in model compression, edge computing, and efficient data representations are being explored, but ensuring high-fidelity semantic understanding under tight latency requirements continues to be a major hurdle for real-world deployment.

**Noise and uncertainty.** Handling noise and uncertainty is a fundamental challenge in semantic map building, particularly when operating in real-world environments characterized by sensor imperfections, dynamic changes and ambiguous semantics. Noise from cameras, LiDARs, or depth sensors can lead to inaccurate observations, while errors in object detection, segmentation, or classification models may introduce semantic inconsistencies in the map. For example, a robot navigating a cluttered indoor environment might repeatedly misclassify similar-looking objects (mistaking a stool for a chair) or fail to detect partially occluded objects or objects in poor lighting. Over time, these errors can compound, leading to semantic drift or contradictions in the map. Furthermore, the uncertainty in sensor data and learned representations is often not explicitly modeled or propagated, which limits the system's ability to reason about the confidence of its predictions. This becomes especially critical in safety applications where overconfident yet incorrect map entries can mislead planning and decision-making. While some approaches use Bayesian filters or probabilistic representations to quantify uncertainty in robotics (Blanco et al., 2008a; Georgakis et al., 2022b), these are computationally expensive and may not scale well. Moreover, semantic mapping methods in embodied AI often assume noiseless sensors in simulated environments, and thus do not transfer well to real robots. Thus, building robust semantic maps that explicitly model and manage uncertainty remains a key open problem in both robotics and embodied AI.

**Multi-modal fusion.** Recent semantic map building approaches increasingly rely on fusing diverse sensory inputs, including vision, depth, audio, natural language, speech etc., to construct rich interpretable representations of the environment. However, robust multi-modal fusion remains a core challenge, with alignment and integration between the modalities a non-trivial problem. For instance, visual inputs provide spatial and appearance cues, while natural language and speech often provide high-level semantic cues that may

be ambiguous, indirect or context-dependent (e.g., 'go to the room where the baby is sleeping'). Moreover such grounding is often compounded by noisy or incomplete observations. Audio cues such as footsteps, conversations or object sounds offer valuable environmental context but are transient and hard to spatially localize or persistently encode. Moreover, multi-modal fusion needs to be done in a way that supports downstream reasoning and generalization in unseen environments. For instance, enabling agents to query the semantic map for complex spatial reasoning (e.g. 'the cup behind the fruit bowl', 'the vase between the kettle and the dish') and affordances (e.g. 'where can I sit', 'which objects are fragile'). The emergence of multi-modal foundation models (Alayrac et al., 2022; Driess et al., 2023; OpenAI, 2023) offers new promise in addressing some of these challenges. These models are capable of aligning and reasoning across modalities, enabling semantic parsing of complex instructions, visual understanding and basic spatial reasoning. However, they are not trained explicitly for map construction or long-term memory integration, limiting their direct applicability in real-time embodied settings. Moreover, deploying such large models on robot hardware poses computational and energy constraints, particularly in latency-sensitive domains like autonomous driving. Thus the challenge lies in building efficient multi-modal fusion strategies that also support reliable and flexible querying, making this an open research problem.

**Lifelong learning.** Lifelong learning poses significant challenges for semantic mapping, particularly in real-world applications where robots are expected to operate continuously over extended periods in dynamic environments. Unlike static maps that assume a fixed world, real-world settings are inherently non-stationary (furniture is rearranged, objects are moved, and scenes evolve with time and seasons). To support lifelong learning and long-term autonomy, semantic maps must be capable of continuous adaptation, i.e. updating outdated information, integrating new observations and distinguishing between transient and persistent changes. This requires temporal reasoning, mechanisms to prevent catastrophic forgetting and the ability to resolve conflicting data collected across different times or viewpoints. The importance of long-term autonomy lies in its potential for practical deployment, i.e. robots that can adapt to changing homes, warehouses or outdoor environments without manual reconfiguration. It is a foundational requirement for truly intelligent agents that not only act within their world but grow and evolve alongside it, building richer and more useful semantic representations over time. However, most current systems are not yet equipped with robust mechanisms for continuous semantic learning and remain limited in handling evolving, open-world environments.

**Standardized evaluation framework.** A significant challenge in semantic map building lies in the absence of standardized evaluation frameworks that can consistently benchmark performance across diverse tasks, environments and map representations, as we discuss at length in Sec. 6.2. While the community has focused more on task-specific extrinsic evaluation, limited progress have been made towards developing intrinsic measures to evaluate the map quality. This lack of standardized intrinsic evaluation hinders progress, as it becomes unclear whether improvements in downstream task success stem from better semantic understanding or other factors like better control policies. Developing general, interpretable and task-agnostic evaluation metrics is crucial, not only to track progress but also to facilitate cross-domain generalization. As open-vocabulary multi-modal queryable general-purpose semantic maps become increasingly central to reasoning in embodied AI and real-world robots, creating a unified benchmark suite that reflects both structural and semantic fidelity is essential for advancing the field.

**Summary.** Several critical challenges remain unaddressed in semantic mapping approaches across various dimensions. Key open problems include building efficient, scalable and real-time maps, which are robust to noise and uncertainty. Building multi-modal maps, that are reliable and flexible, too remains an open challenge. Maps that are able to handle lifelong learning also remain under-explored. Moreover, lack of a standardized evaluation framework require further research and innovation.

## 8 Future research direction

In this section we highlight the current challenges in semantic map building and outline potential directions for future research. Although semantic map building has advanced significantly over the past decades, several challenges and opportunities for improvement remain. The field is evolving toward creating maps that are flexible, general-purpose, open-vocabulary and queryable, enabling the same map representation to support

a wide range of downstream tasks. This shift aims to make maps more versatile and suitable for complex, multi-task robotic systems. Moreover, to enable efficient reasoning about the spatial and semantic structure of the environment, the focus is also on developing dense, scalable, and memory-efficient maps. Such maps should maintain high resolution and detailed spatial information while being computationally efficient and consistent across dynamic and large-scale environments. Achieving this balance is critical for applications that require real-time processing or operate in resource-constrained settings. Furthermore, the emphasis has largely been on evaluating agent performance in downstream tasks using various metrics, with limited attention given to assessing the quality of the built maps. It is however essential to evaluate semantic maps beyond their utility for specific tasks, focusing on metrics such as accuracy, completeness, consistency, and robustness to ensure they are reliable and effective for broader applications. Next, we provide an in-depth discussion of potential future directions that we believe are most critical for advancing research in semantic mapping.

### 8.1 General-purpose maps

Creating general-purpose semantic maps in robotics and embodied AI is crucial for enabling robots to perform a wide variety of tasks in diverse environments with minimal reconfiguration. The idea is to design a general-purpose semantic map that serves as a single comprehensive representation of the environment, combining spatial geometry and semantic information. This eliminates the need for task-specific maps, making it easier to reuse the same map for different tasks such as navigation, object manipulation, and scene understanding. To enable this, the maps need to be open-vocabulary that allow the robots to understand and integrate previously unseen objects using natural language descriptions. This capability broadens the scope of the downstream tasks the robot can perform, especially in unstructured or novel environments. Such semantic maps provide an opportunity to thoroughly evaluate their ability to handle textual queries involving complex spatial and semantic reasoning. Despite the recent progress towards achieving this, it still remains an open research problem. Open-vocabulary maps are currently limited by the pretrained class-agnostic object detectors used in building such maps. For example, these detectors often struggle with detecting small, thin or obscure objects, thus limiting the semantic maps that rely on them. Moreover, open-vocabulary object detectors that incorporate unseen classes through textual descriptions are still not perfect and lack robustness. Thus improving open-vocabulary object detectors that can recognize new objects without extensive retraining could improve the quality of semantic maps that rely on them and hence present a future research avenue. Another challenge that general-purpose map building face is that they are computationally expensive and memory-intensive due to the rich semantic and geometric data stored in them. Balancing map detail with resource efficiency is a challenging task and impacts the ability of robots to process large areas or continuously update the map in real-time without overwhelming computational resources. This presents a future research direction worth pursuing.

### 8.2 Dense yet efficient maps

Following our discussion on various map structures, we find it crucial that the semantic map be able to capture dense visual cues to allow for complex spatial reasoning among objects. For example, beyond addressing straightforward queries like 'Where is the table?', the semantic maps can be assessed on more intricate spatial reasoning queries such as 'Can you retrieve my phone from my desk beside my laptop?'. To perform such reasoning, the semantic map needs to capture fine-grained detail about the spatial arrangement of the objects (phone and laptop on the table) in the scene. While at one end of the spectrum topological maps are too sparse failing to capture the dense semantics of a scene, at the other end the dense geometric maps are too dense capturing redundant empty space information. Although spatial top-down 2D maps exist somewhere in the middle, they fail to capture the semantic information in the third dimension (height). This is crucial where the navigation is in 3D space, for example in drones. Hence there is still a need of a dense-enough map to capture the 3D space in its entirety at the same time intelligently ignoring empty space information. Additionally, a dense map will consume more memory and will be difficult to update. So a dense map representation which is still scalable, memory and computation efficient is a research direction worthwhile to pursue.

### 8.3 Dynamic maps

Current mapping techniques in indoor environments assume that the objects present in the environment are static and only the agent is moving. Although this assumption is reasonable in an indoor environment, it is unrealistic in an outdoor setting in the presence of moving vehicles and people. This entails investigating how well current map building approaches capture moving objects effectively in a dynamic environment and focus on building efficient dynamic maps. Building such maps involves continuous tracking and updating of objects in real-time, as they may move unpredictably. Sensor fusion, which integrates data from multiple sensors like LiDAR and cameras, is often used to detect and track these objects. However, real-time updates can be computationally intensive, particularly in high-traffic areas. Thus, efficiently storing and representing dynamic data in a way that is both memory and computationally scalable remains a key challenge and an ongoing area of research. Moreover, the dynamic nature of these maps complicates their use in downstream tasks. For instance, a robot may need to navigate around a moving pedestrian or vehicle, which requires understanding the object's trajectory and predicting its future movements to avoid collisions. Efficiently integrating this dynamic data into decision-making processes is a significant challenge in autonomous navigation.

### 8.4 Hybrid map structure

A spatial map is able to capture the geometry of a 3D space, which helps to reason about complex spatial relations among objects and areas. A topological map, on the other hand, lacks such geometric spatial understanding but is able to explicitly capture semantic relationships (edges) among objects (nodes). Since both structures have different merits, there have been research to explore a 'hybrid' map structure that leverages the geometric accuracy of spatial maps with the semantic and relational power of topological maps, providing a more comprehensive and efficient tool for complex reasoning tasks. For example, in a large-scale outdoor environment, a robot could use the topological map for long-range navigation to reach one building from another, while switching to a spatial map for close-range navigation to avoid obstacles or interact with objects in a room. A hybrid approach can also help balance the computational load. Topological maps are less resource-intensive and can provide high-level guidance, while spatial maps can be used for precise actions in local regions of interest, reducing the need for continuous, high-cost processing across the entire environment. Moreover, maintaining a separate level of hierarchy to track dynamic objects can also reduce computation load of frequent real-time updates. Although several approaches to hybrid mapping have been proposed in recent years, one of the main challenges in creating hybrid maps is effectively integrating both spatial and topological representations without sacrificing the quality of either. Moreover, combining the two representations requires the robot to make intelligent decisions about when to transition from one to another. Hence significant research is still needed in optimizing hybrid map building, ensuring scalability for large-scale, real-time applications and intelligent algorithms to transition between the different maps.

### 8.5 Devising evaluation metrics

As we discuss in Sec. 6, the evaluating semantic maps in embodied AI research has received limited attention compared to assessing agent performance in downstream tasks. However, we believe that advancing the field requires a stronger emphasis on map evaluation using metrics such as accuracy, completeness, consistency, and robustness. Regardless of the downstream task, maps should be assessed on how well they capture semantic information (accuracy), their geometric and semantic coverage (completeness), their spatial and semantic reliability in dynamic environments (consistency), and their confidence and ability to handle uncertainty and noise (robustness). Establishing standardized evaluation metrics and frameworks for semantic maps remains a critical challenge with substantial opportunities for future research.

### 8.6 Summary

This section outlines promising future directions, as derived from the current challenges in semantic mapping in Sec. 7. As the field moves toward creating flexible, general-purpose, and queryable maps, researchers face open problems in balancing semantic richness with efficiency, ensuring scalability and robustness in dynamic

environments, and developing better evaluation metrics that go beyond task performance to assess map quality directly. These provide a multitude of future research directions, which will help advance the field.

## 9 Conclusion

In this survey, we review a wide range of semantic map-building approaches and categorize them based on their underlying map structure, such as spatial grids, topological graphs, dense geometric and hybrid representations, as well as the nature of semantic encoding, such as explicit or implicit. This perspective is timely in the light of recent advances in foundation models and the increasing need for general-purpose, multi-modal, open-vocabulary and queryable map representations. This survey helps identify key challenges in current semantic mapping paradigms and present promising directions for future research in semantic mapping.

Existing works primarily employ spatial maps to capture geometric layouts or topological maps to model landmark-based relationships. While many robotics studies have explored hybrid maps that combine spatial and landmark information, this approach remains under-explored in embodied AI research. It presents a promising avenue for future work, particularly in leveraging such maps to enhance performance on complex spatial reasoning tasks. We also discuss how dense geometric maps, created by associating semantic information with point clouds, triangle meshes or surfels, offer potential for embodied AI tasks but remain under-explored. While promising for spatial reasoning, their high memory demands, computational inefficiency, and the presence of mostly empty 3D space in indoor environments pose significant challenges. Moreover, most current mapping approaches assume static environments, limiting their applicability in dynamic settings, for example in scenes where furniture is rearranged or humans move around.

We hope that the insights presented in this survey serve to guide and inspire further advancements in the field by the research community.

**Acknowledgements** The authors were supported by a Canada CIFAR AI Chair grant and an NSERC Discovery Grant. We also thank Manolis Savva, Yasutaka Furukawa, Tommaso Campari, Austin T. Wang, Xingguang Yan, Bernadette Bucher, Duy Ta and Sachini Herath, and the anonymous reviewers for their valuable feedback on our paper.

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
