# OpenReview forum: "Semantic Mapping in Indoor Embodied AI - A Survey on Advances, Challenges, and Future Directions"
_TMLR — Accepted by TMLR_

### Review · Reviewer_d4MD · 2025-04-07

**Summary Of Contributions:**

This paper presents a survey of semantic mapping and navigation methods in indoor (simulated) environments. The survey is split into 6 sections: (1) background material on tasks, approaches, and SLAM, (2) the what and how of semantic mapping, (3) an overview of structural representations for maps like grids, topological, and hybrid maps, (4) a summary on map encoding methods, (5) evaluation metrics, and (6) future directions for the field. The survey covers a broad range of works from the embodied-AI community, with a strong emphasis on indoor environments set in simulated household scenes.

**Audience:**

Yes

**Broader Impact Concerns:**

No major concerns.

**Claims And Evidence:**

Yes

**Requested Changes:**

Summarizing from above:
- Improve the connection between introduction and subsections.
- Choose a better example for Figure 1 that is more aligned with “semantic mapping”, rather than
“language grounding for object-navigation”.
- Consider an overview figure/table that summarizes all the categorizations.
- Explicitly distinguish between “semantic mapping” and “object-navigation”. Or separate “object-navigation” into an independent Applications section.
- Explain evaluation metrics in more detail so that the section is self-sufficient.
- Discuss challenges explicitly in a separate section, or make the challenges in the existing sections clear.
- Add additional references provided by AC, and sort them with the existing categories.

**Strengths And Weaknesses:**

Strengths
+ The survey covers a wide range of works in semantic mapping and navigation, which are timely topics in embodied-AI. Quite a lot of learning-based methods struggle to solve long-horizon and partially-observable instruction-following tasks. Semantic mapping provides a structured alternative to end-to-end learning, and this survey provides a nice summary of such methods.
+ The background material on end-to-end vs. modular methods, SLAM, and semantic mapping provide a good intro to keep the survey self-sufficient.
+ The categorization of methods seem appropriate, i.e. map structure into: grid, topological, point-cloud hybrid maps, and map encoding into: explicit and implicit.

Weaknesses
- While the survey is informative, the general structure and writing can be greatly improved. More specifically:
(1) The introduction seems disconnected from the rest of the survey. The first few paragraphs motivate a generic problem in object navigation (i.e., finding “a red and blue striped zebra toy in the nursery”). The same problem is also highlighted in Figure 1. The “zebra toy” example gives the impression that the survey is about vision-language grounding or vision-language navigation, which are potential “applications of semantic mapping” but not about “semantic mapping” itself.  Understandably, object-navigation is closely linked to semantic mapping, but the introduction should be clearly on what is the key problem.
(2) The introduction also doesn’t motivate the key categorizations in the survey, e.g. map structure into: grid, topological etc. It would be good to motivate the categorizations early on, since they are the core contributions of a survey paper.
(3) The survey would greatly benefit from a unifying table or figure to summarize all method subsections and their categories. Right now, it’s difficult to quickly grasp the structure without looking through 26 pages of content.



- The evaluation metrics are mentioned in passing without sufficient detail. For example, SPL is mentioned, without explaining what SPL stands for, i.e. success-weighted path length, or why SPL is necessary. It is important to keep such details self-contained. Besides just summarizing techniques, a survey should also motivate the reason behind techniques, and expound on their strengths and limitations.

- The title mentions “advances, challenges, and future directions”, but “challenges” are not explicitly discussed for all methods. Some challenges are discussed alongside specific methods, but there are no overarching challenges discussed for a group/category of methods.

- The citations in Table 2, 4, and 5 are too sparse and biased towards recent works. The AC has provided a broad set of relevant papers in semantic and topological mapping. It would be good to incorporate these works and sort them into the existing categorization.


Small nits:
- Table 2: the split between Explicit (no semantics), Explicit (semantics), and Implicit to Explicit rows is not clear. Perhaps adding additional lines or separate colors could make it easier to read.

- Some sub-sections end with a summary (e.g. Section 4), but not others. Why not include summaries for all sections?

---

> ### Author Response · Authors · 2025-06-14
>
> We thank the reviewer for their constructive and detailed feedback, as well as for recognizing the value of this survey. We have carefully incorporated all suggestions into our latest revision. Below, we list down our responses and underscore the revisions for each review. We also highlight all the changes in the manuscript with a different color (blue).
>
> 1. We have revised the introduction to connect it to the rest of the survey better and motivated semantic mapping. We have moved (previous) Figure 1 to be Figure 2, which serves as the motivation of semantic mapping. We discuss object navigation separately with all other embodied AI tasks/applications in (current) section 2.1.2.
> >(1) The introduction seems disconnected from the rest of the survey. The first few paragraphs motivate a generic problem in object navigation (i.e., finding “a red and blue striped zebra toy in the nursery”). The same problem is also highlighted in Figure 1. The “zebra toy” example gives the impression that the survey is about vision-language grounding or vision-language navigation, which are potential “applications of semantic mapping” but not about “semantic mapping” itself. Understandably, object-navigation is closely linked to semantic mapping, but the introduction should be clearly on what is the key problem.
>
> 2. We have revised the introduction to motivate the key categorizations done in the paper.
> >(2) The introduction also doesn’t motivate the key categorizations in the survey, e.g. map structure into: grid, topological etc. It would be good to motivate the categorizations early on, since they are the core contributions of a survey paper.
>
> 3. We have moved (previous) Figure 4 to be Figure 1.  It serves as the summary figure of our survey, showing how we categorize the works by the map structure and encoding . We also moved (previous) Table 2 to be Table 1 which serves as a unifying table of all the papers that we survey.  Finally, a short summary of what is covered in the different sections is provided at the end of the introduction.
> >(3) The survey would greatly benefit from a unifying table or figure to summarize all method subsections and their categories. Right now, it’s difficult to quickly grasp the structure without looking through 26 pages of content.
>
> 4. We have revised the Evaluation (current section 6) to contain two sub-sections, one for **Extrinsic** or task-level evaluation (section 6.1) and another for **Intrinsic** or map-level evaluation (section 6.2). In section 6.1, we elaborated on the various metrics (SPL, etc.). We added Table 7 to summarize the extrinsic and intrinsic metrics. We also highlighted the key takeaways in section 6.3 (Summary) and discussed their strengths and limitations.
> >The evaluation metrics are mentioned in passing without sufficient detail. For example, SPL is mentioned, without explaining what SPL stands for, i.e. success-weighted path length, or why SPL is necessary. It is important to keep such details self-contained. Besides just summarizing techniques, a survey should also motivate the reason behind techniques, and expound on their strengths and limitations.
>
> 5. We have added a new section to discuss the 'challenges' in the (current) section 7, where we highlight the key open problems in the field as a whole, across multiple dimensions.
> >The title mentions “advances, challenges, and future directions”, but “challenges” are not explicitly discussed for all methods. Some challenges are discussed alongside specific methods, but there are no overarching challenges discussed for a group/category of methods.
>
> 6. We have included more citations in the current Table 1 (previous Table 2) and added the robotics papers (suggested by the AE as well as all the reviewers) from the past. This table now serves as the summary table for the survey. We also revised current Tables 3,4,6 (previous Tables 3,4,5) to include those citations according to the appropriate classification.
> >The citations in Table 2, 4, and 5 are too sparse and biased towards recent works. The AC has provided a broad set of relevant papers in semantic and topological mapping. It would be good to incorporate these works and sort them into the existing categorization.
>
> 7. We have added horizontal lines in the current Table 1 (previous Table 2) to make the distinction clear.
> >Table 2: the split between Explicit (no semantics), Explicit (semantics), and Implicit to Explicit rows is not clear. Perhaps adding additional lines or separate colors could make it easier to read.
>
> 8. We have added summaries for sections 5 (5.1, 5.2.1, 5.2.2), 6, 7, 8.
> >Some sub-sections end with a summary (e.g. Section 4), but not others. Why not include summaries for all sections?

---

### Review · Reviewer_oQ6x · 2025-04-15

**Summary Of Contributions:**

This is a review paper on semantic mapping in indoor environments for robotics applications. The paper goes over the general setting for  mapping and localization and its motivation in robotics applications (e.g. end to end approaches), goes over some of the methods and points to future directions. The main focus seems to be on the semantic mapping ideas, which is explained phenomenonologically, and broken down into commonly used mapping types (e.g. topological map).

**Audience:**

Yes

**Claims And Evidence:**

Yes

**Requested Changes:**

Please cover transformer based approaches in more detail, together with NeRF and Gaussian splatting. I think it at least needs a passing mention of how transformers can be used for the task - an encoder maps the data to a latent representation, and on the decoder side, we could use it for map/lane estimation tasks e.g. [1]

[1]: https://arxiv.org/abs/2312.16108

**Strengths And Weaknesses:**

+ I liked the conceptual presentation of the work, explaining the process in terms of building blocks. It more or less walks the reader through the the feature extraction process, lifting to 3D and generation of map in world coordinates or BEV, together with SLAM.

- As I work in autonomous driving and BEV fusion, and mostly using transformers as machinery, I feel that the work does not cover this aspect adequately. DETR based approaches should be covered in much more detail. Likewise, NERF based approaches also warrant similar comment.

- The above also goes into covering end to end approaches for planning (i.e. going all the way from perception to trajectories and planning).

- SLAM also needs more coverage with equations.

---

> ### Author Response · Authors · 2025-06-14
>
> We thank the reviewer for their thoughtful feedback, and for acknowledging the significance of our survey. We have carefully incorporated all suggestions into our latest revision. Below, we list down our responses and underscore the revisions for each review. We also highlight all the changes in the manuscript with a different color (blue).
>
> 1. We have included a brief discussion on BEV representations that are popular in autonomous driving and how they are trained using transformers in section 2.1.1. We also included the reference in Table 1. We discuss neural fields and NeRF based approaches in section 4.3.2.
> >As I work in autonomous driving and BEV fusion, and mostly using transformers as machinery, I feel that the work does not cover this aspect adequately. DETR based approaches should be covered in much more detail. Likewise, NERF based approaches also warrant similar comment.
>
> 2. We have included a discussion about end-to-end trained approaches in autonomous driving and planning in section 2.1.1.
> >The above also goes into covering end to end approaches for planning (i.e. going all the way from perception to trajectories and planning).
>
> 3. We have added more details in SLAM (section 2.2), and covered the basic equations in section 2.2.1.
> >SLAM also needs more coverage with equations.
>
> 4. In sections 2.1.1 and 2.2.1, we briefly cover end-to-end transformer-based approaches and BEV representations in autonomous driving and planning. We also discuss neural fields and NeRF based approaches in section 4.3.2 and point-cloud and Gaussian splatting based approaches in section 4.3.1. We also added the reference suggested by the reviewer among others from autonomous driving and BEV based representations.
> > Please cover transformer based approaches in more detail, together with NeRF and Gaussian splatting. I think it at least needs a passing mention of how transformers can be used for the task - an encoder maps the data to a latent representation, and on the decoder side, we could use it for map/lane estimation tasks e.g. 1

---

### Review · Reviewer_qqo5 · 2025-05-30

**Summary Of Contributions:**

This paper presents a survey on semantic mapping in the research field of indoor embodied AI.
The paper first introduces the background of indoor embodied AI and discusses the basic aspect of semantic mapping.
Then it extensively reviews and comments on the relevant papers in the mentioned area.
These papers are further categorized based on two criteria.
Based on map structure, this paper categorizes them into spatial grid maps, topological maps, and point-cloud maps.
Based on semantic encodings stored on the map, this paper categorizes the papers into explicit encodings and implicit encodings.
Then the paper briefly introduces the types of evaluation performed on the built semantic map.
Finally, the paper discusses several potential future directions for the research community to pursue.

**Audience:**

Yes

**Broader Impact Concerns:**

I didn't find any significant broader impact concerns.

**Claims And Evidence:**

No

**Requested Changes:**

Please address the weaknesses I mentioned in the previous section.

Also, I think this survey probably misses many important works and entire subfields in the literature ([R1-R3] as mentioned above and those mentioned by the Action Editor). Please conduct more extensive research and include them.

**Strengths And Weaknesses:**

Strengths:

* The paper presents a categorization of semantic mapping papers based on two criteria.
* The paper is fluently written and easy to follow.


Weaknesses:

* This paper limits its scope to "semantic mapping in indoor embodiment AI". It assumes perfect localization performance and explicitly excludes papers related to "SLAM" and "semantic mapping in robotics (real-world robots)". While I can appreciate the authors' intent to focus specifically on the mapping component (as illustrated in Fig 3), I still feel the SLAM literature captures a huge portion of semantic mapping and the survey should include how the mapping process has evolved in the semantic SLAM literature. Also, most semantic mapping works that are designed for *real-world robots* can be applied or adapted to embodied AI tasks, and I don't think they should be excluded based on this criterion.
* Some categories are missing and not reviewed by the paper, which mostly corresponds to empty entries in the upper triangle of Table 2. I'm sure there are many works that store explicit semantic encodings on point clouds and topological maps (e.g. [R1]). Why are they not included in this survey?
* While Section 4.3 is a "point-cloud map", it mostly reviews the semantic mapping based on neural fields but not the works that actually use point clouds as 3D representations (e.g. [R2]). I suggest the authors separate the implicit 3D representations (SDF, Neural Field, NeRF, 3DGS; 3DGS-based maps are entirely missing from this survey) as an extra subsection in "Map Structure". Review on point cloud-based maps should be redone. ([R1-R3] are all important works in this subfield but they are not mentioned at all in this survey.)
* The Map Evaluation section is very superficial. Discussion of existing benchmarks, evaluation metrics, and quantitative comparisons should included. A table listing and comparing popular benchmarks and evaluation metrics from different embodied AI tasks is helpful. The quantitative comparisons can come from some existing popular datasets and benchmarks (e.g. 3D semantic segmentation on ScanNet, navigation performance on HM3D dataset). Without these, I hardly learn any takeaways from this paper on how different techniques compare to each other.
* Section 2 is loosely written and I don't see how it provides value to a survey on semantic mapping. Section 3 reads similarly loose. Some less relevant discussions can be omitted.
* Overall the writing of the paper needs to be improved. Specifically, I want the authors to highlight how they "draw new connections, highlight trends, and suggest new problems in an area" (according to the TMLR review guidance).


[R1] J. McCormac, A. Handa, A. Davison and S. Leutenegger, "SemanticFusion: Dense 3D semantic mapping with convolutional neural networks," 2017 IEEE International Conference on Robotics and Automation (ICRA), Singapore, 2017, pp. 4628-4635, doi: 10.1109/ICRA.2017.7989538.

[R2] Jatavallabhula, Krishna Murthy, Alihusein Kuwajerwala, Qiao Gu, Mohd Omama, Tao Chen, Alaa Maalouf, Shuang Li et al. "Conceptfusion: Open-set multimodal 3d mapping." RSS 2023.

[R3] Werby, Abdelrhman, Chenguang Huang, Martin Büchner, Abhinav Valada, and Wolfram Burgard. "Hierarchical Open-Vocabulary 3D Scene Graphs for Language-Grounded Robot Navigation." RSS 2024.

---

> ### Author Response · Authors · 2025-06-14
>
> We thank the reviewer for their constructive and thorough feedback on the survey paper. We have carefully incorporated all suggestions into our latest revision. Below, we list down our responses and underscore the revisions for each review. We also highlight all the changes in the manuscript with a different color (blue).
>
> 1. We have included more works from the SLAM and Semantic SLAM literature in the survey. More specifically, in section 2.2.3 we highlight how mapping has evolved in the semantic SLAM, in Table 1, Table 4, section 4.2, & section 6 we have added more references from SLAM and semantic SLAM.
> > This paper limits its scope to "semantic mapping in indoor embodiment AI". It assumes perfect localization performance and explicitly excludes papers related to "SLAM" and "semantic mapping in robotics (real-world robots)". While I can appreciate the authors' intent to focus specifically on the mapping component (as illustrated in Fig 3), I still feel the SLAM literature captures a huge portion of semantic mapping and the survey should include how the mapping process has evolved in the semantic SLAM literature. Also, most semantic mapping works that are designed for real-world robots can be applied or adapted to embodied AI tasks, and I don't think they should be excluded based on this criterion.
>
> 2. We have added more references in Table 1 (previous Table 2) and reviewed them in appropriate sections, including all references mentioned in this review as well as other reviews and the suggestions from the AE.
> > Some categories are missing and not reviewed by the paper, which mostly corresponds to empty entries in the upper triangle of Table 2. I'm sure there are many works that store explicit semantic encodings on point clouds and topological maps (e.g. [R1]). Why are they not included in this survey?
>
> 3. We have revised section 4.3 to separate 'point-cloud map' (section 4.3.1) from 'neural fields' (section 4.3.2) and discuss these under heading 'dense geometric map'. In section 4.3.2, we have included prior works on SDF, Neural fields, NeRF and 3DSG based maps. We discuss how prior works in robotics, embodied AI and 3D scene understanding use both these types of maps and added Table 5 to summarize them.
> > While Section 4.3 is a "point-cloud map", it mostly reviews the semantic mapping based on neural fields but not the works that actually use point clouds as 3D representations (e.g. [R2]). I suggest the authors separate the implicit 3D representations (SDF, Neural Field, NeRF, 3DGS; 3DGS-based maps are entirely missing from this survey) as an extra subsection in "Map Structure". Review on point cloud-based maps should be redone. ([R1-R3] are all important works in this subfield but they are not mentioned at all in this survey.)
>
> 4. We have revised the Evaluation (current section 6) to contain two sub-sections, one for Extrinsic or task-level evaluation (section 6.1) and another for Intrinsic or map-level evaluation (section 6.2). We added Table 7 to summarize the extrinsic and intrinsic metrics. We also highlighted the key takeaways in section 6.3 (Summary) and discussed their strengths and limitations.
> > The Map Evaluation section is very superficial. Discussion of existing benchmarks, evaluation metrics, and quantitative comparisons should included. A table listing and comparing popular benchmarks and evaluation metrics from different embodied AI tasks is helpful.
>
> 5. However, we have decided not to include the quantitative results for existing benchmarks in section 6.1 because (a) our aim for the Evaluation section is to highlight the gap in intrinsic evaluation of the map itself, (b) to motivate future research in intrinsic evaluation, and (c) extrinsic or task-level evaluation is extremely dependent upon the task and is often coupled with other methodological decisions (e.g. navigation performance on HM3D can heavily depend on the navigation policy as well as the map) and it is beyond the scope of this work to cover all the different tasks and impact of different component choices. For embodied AI task performance, we direct the readers to this survey [Deitke et al. Retrospectives on the embodied AI workshop. 2022.].
> >The quantitative comparisons can come from some existing popular datasets and benchmarks (e.g. 3D semantic segmentation on ScanNet, navigation performance on HM3D dataset). Without these, I hardly learn any takeaways from this paper on how different techniques compare to each other.
>
> [contd.]

---

> > ### Author Response · Authors · 2025-06-14
> >
> > [contd.]
> >
> > 6. We have revised section 2 to connect the background reading to the rest of the survey. We have also revised section 3 to be more structured and relevant to the rest of the survey, and removed the redundant section 3.5 (moved some of the relevant discussions to section 7 - challenges).
> > >Section 2 is loosely written and I don't see how it provides value to a survey on semantic mapping. Section 3 reads similarly loose. Some less relevant discussions can be omitted.
> >
> > 7. We have now improved the introduction (section 1) - (a) differentiated this survey from other surveys (draw new connections), (b) highlighted how recent advances in foundation models have shaped the semantic mapping field (highlight trends), and (c) identified challenges and suggested future research directions (suggest new problems in an area).
> > >Overall the writing of the paper needs to be improved. Specifically, I want the authors to highlight how they "draw new connections, highlight trends, and suggest new problems in an area" (according to the TMLR review guidance).

---

### Comment · Action_Editor_izDm · 2025-03-12
**Missing large body of related work**

The paper is missing a **large** body of work semantic mapping [1,8,9,11,13,14,16,21,22,24,25,30,31,32,34] and work closely related to the field [2,3,4,5,6,7,10,12,15,17,18,19,20,23,26,27,28,29,33]. Given that the paper is intending to provide a survey on the field, it is critical that it provide a broader discussion of the extensive work that has been done on and related to semantic mapping in robotics settings.

[1] Arkin, J., Park, D., Roy, S., Walter, M. R., Roy, N., Howard, T. M., and Paul, R. (2020). Multimodal estimation and communication of latent semantic knowledge for robust execution of robot instructions. International Journal of Robotics Research, 39(10–11):1279–1304.

[2] Arkin, J., Paul, R., Park, D., Roy, S., Roy, N., and Howard, T. M. (2018). Real-time human-robot communication for manipulation tasks in partially observed environments. In Proceedings of the International Symposium on Experimental Robotics (ISER), pages 448–460.

[3] Aydemir, A., Pronobis, A., G¨obelbecker, M., and Jensfelt, P. (2013). Active visual object search in unknown environments using uncertain semantics. IEEE Transactions on Robotics, 29(4):986–1002.

[4] Aydemir, A., Sj¨o¨o, K., Folkesson, J., Pronobis, A., and Jensfelt, P. (2011). Search in the real world: Active visual object search based on spatial relations. In Proceedings of the IEEE International Conference on Robotics and Automation (ICRA), pages 2818–2824

[5] Bansal, S., Tolani, V., Gupta, S., Malik, J., and Tomlin, C. (2019). Combining optimal control and learning for visual navigation in novel environments. In Proceedings of the Conference on Robot Learning (CoRL), pages 420–429.

[6] Chen, D. L. and Mooney, R. J. (2011). Learning to interpret natural language navigation instructions from observations. In Proceedings of the National Conference on Artificial Intelligence (AAAI), pages 859–865.

[7] Duvallet, F., Kollar, T., and Stentz, A. (2013). Imitation learning for natural language direction following through unknown environments. In Proceedings of the IEEE International Conference on Robotics and Automation (ICRA), pages 1047–1053.

[8] Duvallet, F., Walter, M. R., Howard, T., Hemachandra, S., Oh, J., Teller, S., Roy, N., and Stentz, A. (2014). Inferring maps and behaviors from natural language instructions. In Proceedings of the International Symposium on Experimental Robotics (ISER).

[9] Galindo, C., Saffiotti, A., Coradeschi, S., Buschka, P., Fernandez-Madrigal, J., and Gonzalez, J. (2005). Multi-hierarchical semantic maps for mobile robotics. In Proceedings of the IEEE/RSJ International Conference on Intelligent Robots and Systems (IROS), pages 2278–2283.

[10] Gong, Z. and Zhang, Y. (2018). Temporal spatial inverse semantics for robots communicating with humans. In Proc. IEEE Int’l Conf. on Robotics and Automation (ICRA).

[11] Hemachandra, S., Duvallet, F., Howard, T. M., Roy, N., Stentz, A., and Walter, M. R. (2015). Learning models for following natural language directions in unknown environments. In Proceedings of the IEEE International Conference on Robotics and Automation (ICRA), pages 5608–5615.

[12] Hemachandra, S., Kollar, T., Roy, N., and Teller, S. (2011). Following and interpreting narrated guided tours. In Proceedings of the IEEE International Conference on Robotics and Automation (ICRA), pages 2574–2579.

[13] Hemachandra, S., Walter, M. R., Tellex, S., and Teller, S. (2014). Learning spatial-semantic representations from natural language descriptions and scene classifications. In Proceedings of the IEEE International Conference on Robotics and Automation (ICRA), pages 2623–2630.

[14] Landsiedel, C., Rieser, V., Walter, M. R., and Wollherr, D. (2017). A review of spatial reasoning and interaction for real-world robotics. Advanced Robotics, 31(5):222–242.

[15] MacMahon, M., Stankiewicz, B., and Kuipers, B. (2006). Walk the talk: Connecting language, knowledge, and action in route instructions. In Proceedings of the National Conference on Artificial Intelligence (AAAI), pages 1475–1482.

[16] Martınez Mozos, O., Triebel, R., Jensfelt, P., Rottmann, A., and Burgard, W. (2007). Supervised semantic labeling of places using information extracted from sensor data. Robotics and Autonomous Systems, 55(5):391–402.

[17] Matuszek, C., Fox, D., and Koscher, K. (2010). Following directions using statistical machine translation. In Proceedings of the ACM/IEEE International Conference on Human-Robot Interaction (HRI), pages 251–258.

---

> ### Comment · Action_Editor_izDm · 2025-03-12
>
> [18] Matuszek, C., Herbst, E., Zettlemoyer, L., and Fox, D. (2012). Learning to parse natural language commands to a robot control system. In Proceedings of the International Symposium on Experimental Robotics (ISER), pages 403–415.
>
> [19] Meger, D., Forss´en, P.-E., Lai, K., Helmer, S., McCann, S., Southey, T., Baumann, M., Little, J. J., and Lowe, D. G. (2008). Curious George: An attentive semantic robot. Robotics and Autonomous Systems, 56(6):503–511.
>
> [20] Mei, H., Bansal, M., and Walter, M. R. (2016). Listen, attend, and walk: Neural mapping of navigational instructions to action sequences. In Proceedings of the National Conference on Artificial Intelligence (AAAI).
>
> [21] Patki, S., Daniele, A., Walter, M., and Howard, T. (2019). Inferring compact representations for efficient natural language understanding of robot instructions. In Proceedings of the IEEE International Conference on Robotics and Automation (ICRA), pages 6926–6933.
>
> [22] Patki, S., Fahnestock, E., Howard, T. M., and Walter, M. R. (2020). Language-guided semantic mapping and mobile manipulation in partially observable environments. In Proceedings of the Conference on Robot Learning (CoRL), pages 1201–1210.
>
> [23] Paul, R., Arkin, J., Aksaray, D., Roy, N., and Howard, T. M. (2018). Efficient grounding of abstract spatial concepts for natural language interaction with robot platforms. International Journal of Robotics Research, 37(10):1269–1299.
>
> [24] Pronobis, A. and Jensfelt, P. (2012). Large-scale semantic mapping and reasoning with heterogeneous modalities. In Proceedings of the IEEE International Conference on Robotics and Automation (ICRA), pages 3515–3522.
>
> [25] Pronobis, A., Martınez Mozos, O., Caputo, B., and Jensfelt, P. (2010). Multi-modal semantic place classifi- cation. International Journal of Robotics Research, 29(2–3):298–320.
>
> [26] Tai, L., Paolo, G., and Liu, M. (2017). Virtual-to-real deep reinforcement learning: Continuous control of mobile robots for mapless navigation. In Proceedings of the IEEE/RSJ International Conference on Intelligent Robots and Systems (IROS), pages 31–36.
>
> [27] Tellex, S., Kollar, T., Dickerson, S., Walter, M. R., Banerjee, A. G., Teller, S., and Roy, N. (2011). Understanding natural language commands for robotic navigation and mobile manipulation. In Proceedings of the National Conference on Artificial Intelligence (AAAI), pages 1507–1514.
>
> [28] Thomason, J., Sinapov, J., Mooney, R. J., and Stone, P. (2018). Guiding exploratory behaviors for multimodal grounding of linguistic descriptions. In Proceedings of the National Conference on Artificial Intelligence (AAAI), pages 5520–5527.
>
> [29] Tucker, M., Aksaray, D., Paul, R., Stein, G. J., and Roy, N. (2017). Learning unknown groundings for natural language interaction with mobile robots. In International Symposium on Robotics Research (ISRR), pages 317–333.
>
> [30] Vasudevan, S. and Siegwart, R. (2008). Bayesian space conceptualization and place classification for semantic maps in mobile robotics. Robotics and Autonomous Systems, 56(6):522–537.
>
> [31] Walter, M. R., Hemachandra, S., Homberg, B., Tellex, S., and Teller, S. (2013). Learning semantic maps from natural language descriptions. In Proceedings of Robotics: Science and Systems (RSS).
>
> [32] Walter, M. R., Hemachandra, S., Homberg, B., Tellex, S., and Teller, S. (2014). A framework for learning semantic maps from grounded natural language descriptions. International Journal of Robotics Research, 33(9):1167–1190.
>
> [33] Walter, M. R., Patki, S., Daniele, A.F., Fahnestock, E., Duvallet, F., Hemachandra, S. Oh, J., Stentz, A., Roy, N., and Howard, T.M. (2022). Language understanding for field and service robots in a priori unknown environments. Field Robotics 2:1191-1231.
>
> [34] Zender, H., Mart´ınez Mozos, O., Jensfelt, P., Kruijff, G., and Burgard, W. (2008). Conceptual spatial representations for indoor mobile robots. Robotics and Autonomous Systems, 56(6):493–502.

---

> > ### Author Response · Authors · 2025-03-12
> >
> > We appreciate your feedback on our paper and would like to thank you for providing us with the references. We will include them in the paper. Do you suggest we make the edits now or wait for more reviews?

---

> > > ### Comment · Action_Editor_izDm · 2025-03-12
> > >
> > > Hi,
> > >
> > > Thanks. No, I don't think that you need to make the edits now.
> > >
> > > Best,\
> > > AE

---

> > > > ### Author Response · Authors · 2025-06-14
> > > >
> > > > We sincerely thank the Action Editor for suggesting the list of papers that we missed originally.
> > > >
> > > > 1. We have added all the reference papers suggested in Table 1 and also in appropriate sections.
> > > > 2. We also added the references in appropriate Tables 3,4,6 (previous Tables 3,4,5).
> > > >
> > > > We also highlight all the changes in the manuscript with a different color (blue).

---

### Author Response · Authors · 2025-06-14
**Summary of the revision**

We sincerely thank all the reviewers for their thoughtful and thorough feedback and for acknowledging the significance of this survey. We also thank the Action Editor for suggesting the list of works we missed initially. We have carefully addressed all comments and incorporated the suggested revisions, which we believe have substantially strengthened the manuscript.

We summarize below all the updates that we have incorporated. We also highlight all the changes in the manuscript with a different color (blue).

1. We have revised the introduction to connect it to the rest of the survey better and to motivate the key categorizations.
2. To summarize the survey, we have Figure 1 (previous Figure 4) and Table 1 (previous Table 2). They together serve as the summary of the survey.
3. We have added all the reference papers suggested by the AE and all the reviewers in Table 1 and also in appropriate sections. We also add the references in appropriate Tables 3,4,6 (previous Tables 3,4,5).
4. We have moved (previous) Figure 1 to be Figure 2, which serves as the motivation of semantic mapping.
5. We have revised the Evaluation (current section 6) to contain two sub-sections, one for Extrinsic or task-level evaluation (section 6.1) and another for Intrinsic or map-level evaluation (section 6.2). We added Table 7 to summarize the metrics and highlighted the key takeaways in section 6.3 (Summary) and discussed their strengths and limitations.
6. We have added a new section to discuss the 'challenges' in the (current) section 7, where we highlight the key open problems in the field as a whole, across multiple dimensions.
7. To be uniform with section 4, we have added summaries for all sections 5 (5.1, 5.2.1, 5.2.2), 6, 7, 8.
8. We have revised section 4.3 to have 'dense geometric map', under which we separately discuss 'point-cloud map' (section 4.3.1) and 'neural fields' (section 4.3.2). We discuss how prior works in robotics, embodied AI and 3D scene understanding use both these types of maps and added Table 5 to summarize them.
9. We have revised section 2 to connect the background reading to the rest of the survey.
10. We have also revised section 3 to be more structured and relevant to the rest of the survey, and removed the redundant section 3.5 (moved some of the relevant discussions to section 7 - challenges).

---

### Decision · Action_Editor_izDm · 2025-07-18

**Recommendation:** Accept as is

**Audience:**

Yes

**Audience Explanation:**

As the reviewers note, this paper offers a timely overview of a wide range of research on semantic mapping, which is particularly relevant in light of how semantic maps can help address the difficulties associated with using modern learning-based instruction-following approaches in large environments and for long-duration tasks. With the updates made during the discussion phase to provide a more thorough review of semantic mapping research and to better convey the paper's contribution, it will be of interest to to many in the community.

**Claims And Evidence:**

Yes

**Claims Explanation:**

The paper provides a survey of research on semantic mapping and navigation in indoor environments, with a particular emphasis on simulated household settings. After discussing the general problem of semantic mapping, the survey reviews the different environment (map) representations that are standard in the literature, the nature of the encoding (i.e., implicit vs. explicit), and various evaluation metrics, and then discusses future research directions related to semantic mapping.

As noted below, the reviewers emphasize the value and relevance of a survey on semantic mapping research. However, the reviewers shared concerns about the paper as initially submitted. These included the notable omission of the large-body of work focused on SLAM-based approaches to semantic mapping (i.e., targeting settings where the robot's pose is unknown), an issue that the AE noted as well. Similarly, the reviewers found that the paper was missing relevant work on different semantic map encodings, point cloud-based maps, as well as representations like DETR, NeRF, and Gaussian splatting. Additionally, the reviewers emphasized the need to improve the writing and general structure throughout much of the paper. At least two reviewers emphasized that introduction should better highlight the key contributions of the survey and provide clearer connections with the subsequent discussions that are the focus of the paper. At least two reviewers found that the submission didn't include sufficient discussion or analysis of performance metrics, including the various benchmarks that are typically used to compare algorithms.

In response to the reviewers’ feedback, the authors significantly revised the paper and clearly explained their changes as part of the discussion phase. The reviewers acknowledge these changes in their final recommendation, emphasizing their appreciation for the authors' efforts and noting that the updated paper addresses the key concerns that they had with the paper as originally submitted.